# Hemodynamic and electromechanical effects of paraquat in rat heart

**Chih-Chuan Lin**[1], **Kuang-Hung Hsu**[2], **Chia-Pang Shih**[3], **Gwo-Jyh Chang**[4,5]*

**1** Department of Emergency Medicine, Chang Gung Memorial Hospital, College of Medicine, Chang Gung University, Tao-Yuan, Taiwan, **2** Laboratory for Epidemiology, Department of Health Care Management, and Healthy Aging Research Center, Chang Gung University, Tao-Yuan, Taiwan, **3** Department of Nursing, Yuanpei University of Medical Technology, Hsin-Chu, Taiwan, **4** Graduate Institute of Clinical Medicinal Sciences, College of Medicine, Chang Gung University, Tao-Yuan, Taiwan, **5** Cardiovascular Division of Medicine, Chang Gung Memorial Hospital, Tao-Yuan, Taiwan

* gjchang@mail.cgu.edu.tw

**Data Availability Statement:** All relevant data are within the paper and its Supporting Information files.

**Funding:** This work was supported by Chang Gung Memorial Hospital, Taiwan (grants CMRPG

## Abstract

Paraquat (PQ) is a highly lethal herbicide. Ingestion of large quantities of PQ usually results in cardiovascular collapse and eventual mortality. Recent pieces of evidence indicate possible involvement of oxidative stress- and inflammation-related factors in PQ-induced cardiac toxicity. However, little information exists on the relationship between hemodynamic and cardiac electromechanical effects involved in acute PQ poisoning. The present study investigated the effects of acute PQ exposure on hemodynamics and electrocardiogram (ECG) *in vivo*, left ventricular (LV) pressure in isolated hearts, as well as contractile and intracellular $Ca^{2+}$ properties and ionic currents in ventricular myocytes in a rat model. In anesthetized rats, intravenous PQ administration (100 or 180 mg/kg) induced dose-dependent decreases in heart rate, blood pressure, and cardiac contractility (LV $+dP/dt_{max}$). Furthermore, PQ administration prolonged the PR, QRS, QT, and rate-corrected QT (QTc) intervals. In Langendorff-perfused isolated hearts, PQ (33 or 60 μM) decreased LV pressure and contractility (LV $+dP/dt_{max}$). PQ (10–60 μM) reduced the amplitudes of $Ca^{2+}$ transients and fractional cell shortening in a concentration-dependent manner in isolated ventricular myocytes. Moreover, whole-cell patch-clamp experiments demonstrated that PQ decreased the current amplitude and availability of the transient outward $K^+$ channel ($I_{to}$) and altered its gating kinetics. These results suggest that PQ-induced cardiotoxicity results mainly from diminished $Ca^{2+}$ transients and inhibited $K^+$ channels in cardiomyocytes, which lead to LV contractile force suppression and QTc interval prolongation. These findings should provide novel cues to understand PQ-induced cardiac suppression and electrical disturbances and may aid in the development of new treatment modalities.

## Introduction

Paraquat (PQ) is a highly toxic herbicide that causes significant mortality when ingested. It is a widely used herbicide because of its excellent ability to eliminate weeds [1]. However, intentionally ingesting PQ to commit suicide is a common problem in some Asian countries (e.g.,

381571-3) to C.C. Lin. The funders had no role in study design, data collection, analysis, decision to publish, or manuscript preparation.

**Competing interests:** The authors have declared that no competing interests exist.

Taiwan, Japan, South Korea, Malaysia, and Sri Lanka) [2–5]. No specific and effective therapy for clinical PQ poisoning currently exists. Nevertheless, early detection and immediate treatment commencement for specific organ injuries are crucial for the prognosis of PQ intoxication prognosis.

Fulminant PQ poisoning patients with cardiovascular collapse and multiple organ failure have the highest mortality rate of over 50% [6,7]. Expiry mostly occurs within 24–72 h or one week [6,7]. The heart is one of the major severely injured organs in PQ poisoning. Histologically, acute PQ poisoning may cause PQ accumulation and direct myocardial injury and dysfunction [8–10]. Our previous study indicated that acute PQ poisoning also causes rate-corrected QT (QTc) interval prolongation which may serve as a useful prognostic factor for poor clinical outcome [11]. As known, QTc interval prolongation may predispose subjects to ventricular arrhythmias [12]. Moreover, QTc prolongation is thought to be related to left ventricular (LV) systolic dysfunction in cardiac disease patients [13]. In human PQ poisoning cases, minimal changes on the electrocardiogram (ECG) to extensive myocardial necrosis had been observed [10,14]. Moreover, severe PQ cardiotoxicity may lead to sudden death induced by either cardiac arrhythmias or acute heart failure [10,14]. Several studies have recently demonstrated that inflammation and oxidative stress evoked through pathways involving the activation of Toll-like receptor 4 (TLR4) and endothelin receptor A (ET$_A$) or downregulation of Forkhead box O3 (FoxO3) [15–17] may contribute to PQ-induced cardiac toxicity. It is widely accepted that PQ catalyzes the formation of ROS which may cause lipid peroxidation leading to cardiac dysfunction [18]. However, the hemodynamic and cardiac electromechanical effects following acute PQ intoxication have not been elucidated. Moreover, the precise cellular mechanism underlying the PQ exposure-induced QTc prolongation remains unknown. Therefore, this study aimed to investigate the underlying mechanisms of PQ-induced cardiotoxicity both *in vivo* and *in vitro*. Thus, its impact on hemodynamic and ECG parameters of anesthetized rats; its effects on LV pressure of isolated hearts; as well as its effects on Ca$^{2+}$ transients, cell shortening, and K$^+$ currents of ventricular myocytes have been defined.

## Materials and methods

### Experimental animals

All experiments were approved by the Institutional Animal Care and Use Committee of Chang Gung University (IACUC approval numbers CGU07-16 and CGU09-015) and performed in accordance with the Guide for the Care and Use of Laboratory Animals published by the US National Institutes of Health (NIH publication No. 85–23, revised 1996). Adult male Sprague–Dawley rats, which were healthy and weighed 250–300 g, were purchased from Bio-LASCO Tech. Co. (Taipei, Taiwan). Animals were housed in polypropylene cages and kept under standard laboratory conditions (12/12 h light/darkness, 45–60% relative humidity, and 24–26˚C room temperature) for at least one week before starting the experiments. Food and water were available *ad libitum*. Animal experiments were performed at the same time of the day. Furthermore, 101 animals were included in the study with 30 and 28 animals used for *in vivo* and isolated heart studies, respectively, and 43 for single cardiomyocyte study. All efforts were made to minimize animal suffering.

### Materials

PQ dichloride (Mw 257.16 g/mol) was purchased from Sigma-Aldrich (Cat. No. 856177; St. Louis, MO, USA) and prepared in physiological saline. Fura-2-acetoxymethyl ester (Fura-2-AM) and Pluronic F-127 were purchased from Molecular Probes (Eugene, OR, USA) and dissolved in dimethylsulfoxide. All other drugs were purchased from Sigma-Aldrich and

prepared in physiological saline before the start of each experiment. In clinical situations, PQ poisoning is classified as mild, moderate to severe, and massive [7]. In humans, ingestion of dosages >40 mg/kg BW may cause acute fulminant poisoning with an extremely high mortality rate [7]. The reported $LD_{50}$ value in rats was around 57 mg/kg. In general, the lethal dose in rodents is much higher compared with humans. Therefore, 100 and 180 mg/kg (approximately twice and triple $LD_{50}$, respectively) were used for *in vivo* study to simulate the clinical severity of PQ poisoning. For *in vitro* study, the concentrations used were estimated according to the calculated blood concentrations of PQ, assuming that blood volume (in mL) represents 7% of the body weight (in g). Consequently, dosages of 100 and 180 mg/kg PQ should produce approximate blood concentrations of 5.56 and 10 mM, respectively. However, a much lower concentration (e.g., 1–60 μM) was chosen than the estimated ones for the current *in vitro* study. The action of the drug is confined in the heart and the drug can also easily access its target sites in the *in vitro* conditions.

### Hemodynamic and ECG measurements in anesthetized rats

Adult male Sprague-Dawley rats weighing 250–300 g were anesthetized with urethane (1.25 g/kg, i.p.). To measure arterial pressure, a polyethylene (PE50) cannula filled with heparinized saline (25 IU/mL) was inserted into the femoral artery as previously described [19]. The femoral vein was cannulated with a PE50 catheter for drug or vehicle administration. The arterial cannula was connected to an MLT0380/D pressure transducer (ADInstruments, Bella Vista, Australia) linked to a QuadBridge amplifier (ADInstruments). ECG needles were connected to a biological amplifier (ADInstruments), and lead II ECGs were simultaneously recorded. QT intervals were rate corrected according to normalized Bazett's formula $QTc = QT/(RR/f)^{1/2}$, where $f$ is the normalized factor according to the basal RR duration [20]. In our study, the average RR duration at basal conditions was 182 ms. Consequently, the value of 180 ms was chosen as the $f$ value. A 1.9-F microtip pressure–volume catheter (Model FTS-1912B, Scisense Inc., London, Canada) was advanced from the right carotid artery into the LV chamber under pressure control for measuring LV pressure (LVP). LVP signals were continuously recorded using a pressure–volume conductance system (Model 891A, Scisense). Output signals from these amplifiers were connected to a Ponemah ACQ16 acquisition system (DSI Ponemah, Valley View, OH, USA), recorded at a sampling rate of 4 kHz, and stored and displayed on a computer. All arterial and LV pressure data were analyzed using a data analysis program (P3 Plus4.80-SP4, DSI Ponemah). Consequently, the mean arterial blood pressure (MAP), heart rate (HR), systolic blood pressure (SBP), diastolic blood pressure (DBP), LV end-systolic pressure (LVESP), LV end-diastolic pressure (LVEDP), and maximal rate of rising ($+dP/dt_{max}$) and fall ($-dP/dt_{max}$) of LVP were computed. After 20 min of stabilization, saline or PQ (100 or 180 mg/kg) solution was intravenously administered in the control or drug-treated groups, respectively. The infusion volume was 1 mL/kg and given for 1 min. Following the infusion test, animals were injected with urethane (1.25 g/kg, i.v.) to induce deep anesthesia and then sacrificed by cervical dislocation.

### Intraventricular pressure measurement in Langendorff-perfused rat hearts

The rats were anesthetized with pentobarbital sodium (50 mg/kg, i.p.) and placed on an operating table. Rat hearts were immediately excised, mounted on a Langendorff apparatus, and perfused at a constant pressure of approximately 55 mmHg with oxygenated (95% $O_2$ and 5% $CO_2$) normal Tyrode's solution containing (in mM): NaCl 137.0, KCl 5.4, $MgCl_2$ 1.1, $NaHCO_3$ 11.9, $NaH_2PO_4$ 0.33, $CaCl_2$ 1.8, and dextrose 11.0 at 37°C as previously described [21]. A latex balloon (size No. 5, Radnoti, Monrovia, CA, USA) connected by a short stainless steel tube to a

pressure transducer (P23XL-1, Becton, Dickinson & Co., Franklin Lakes, NJ, USA) was inserted into the LV cavity via the left atrium. The balloon was inflated with 0.04 mL distilled water, sufficient to produce an end-diastolic pressure of 8–12 mmHg. The ventricles were paced electrically at 300 beats per minute by platinum contact electrodes positioned on the right ventricular apex. Data were recorded on a WindowGraf recorder (Gould Inc., Cleveland, OH, USA) and digitized with a computer-based data acquisition system (PowerLab/4SP with Chart 5 software, ADInstruments). Each preparation was allowed to equilibrate for 2–2.5 h before drug testing. LV developed pressure (LVDP) was calculated by subtracting LVEDP from the LV peak systolic pressure. LVP signal differentiation was used to determine LV $+dP/dt_{max}$ and $-dP/dt_{max}$.

## Single cardiac myocyte isolation

Single ventricular myocytes from adult rats were obtained by an enzymatic dissociation method previously described [19,21]. In brief, the excised heart was mounted on a Langendorff apparatus and retrogradely perfused at a rate of 6 mL/min/g cardiac tissue by a peristaltic pump with nominally $Ca^{2+}$-free 4-(2-hydroxyethyl)-1-piperazineethanesulfonic acid (HEPES)-buffered Tyrode's solution containing (in mM): NaCl 137.0, KCl 5.4, $KH_2PO_4$ 1.2, $MgSO_4$ 1.22, dextrose 22.0, and HEPES 6.0, titrated to pH 7.4 with NaOH. The perfusate was oxygenated and maintained at $37 \pm 0.2°C$ by a heating circulator. After 5 min, the perfusate was changed to the same solution containing 0.3 mg/mL collagenase (Type II, Sigma-Aldrich) and 0.1 mg/mL protease (Type XIV, Sigma-Aldrich). After digestion for 7–15 min, the residual enzyme solution was removed by perfusing 0.05 mM $Ca^{2+}$-containing Tyrode's solution. The ventricles were then separated from the atria, dispersed, and stored in 0.2 mM $Ca^{2+}$-containing Tyrode's solution for later use. Rod-shaped $Ca^{2+}$-tolerant viable cells with clear striations were used for experiments.

## Measurements of intracellular $Ca^{2+}$ transients and cell shortening

Ventricular myocytes were loaded with the fluorescent $Ca^{2+}$-sensitive indicator, fura-2, by incubating cells in 0.5 mM $Ca^{2+}$-containing HEPES-buffered solution containing 5 μM fura-2-AM and 2% Pluronic F-127 for 30 min at room temperature, as previously described [19,21]. After washing out the excess fura-2-AM, cells were stored in 0.5 mM $Ca^{2+}$-containing HEPES-buffered solution. Fura-2–loaded myocytes were transferred to 1.8 mM $Ca^{2+}$-containing HEPES buffer for at least 30 min before beginning the experiments. Myocytes were placed on an inverted microscope (Axio Observer Z1, Carl Zeiss MicroImaging GmbH, Jena, Germany) equipped with a heated (37°C) chamber. Myocytes were then electrically stimulated using a pair of platinum electrodes with a 2-ms and twofold threshold rectangular voltage pulse at 1 Hz. The cells were illuminated with ultraviolet light from a light source (DeltaScan, PTI-HORIBA Scientific, Edison, NJ, USA). The excitation lights with a 340 or 380 nm wavelength passed through a ×40 oil immersion objective to the cell by a dichroic mirror. The emission light passed through a filter (510 nm) and was detected by a photomultiplier tube and recorded using a RatioMaster fluorometer (PTI-HORIBA Scientific). Furthermore, excitation light was intermittently applied and attenuated by 90% using a neutral density filter to minimize photobleaching of fura-2. Signals were acquired using a data acquisition system controlled with professional software (FeliX32™, PTI-HORIBA Scientific). Moreover, intracellular $Ca^{2+}$ was directly expressed as the ratio of the light emitted at excitation wavelengths of 340 and 380 nm ($F_{340}/F_{380}$) because the fura-2 ratio is not a linear function of intracellular $Ca^{2+}$ concentration when cells are loaded with fura-2-AM. Background fluorescence measured from a cell-free field was subtracted from all recordings before ratio calculation. Cell

shortening was measured optically with an R12 dual raster line edge detector system (Crescent Electronics, Sandy, UT, USA). Images of contracting myocytes were viewed with a charge-coupled device camera mounted with a dual C port adaptor on the side port of the microscope. The camera signals were linked to the edge detector electronics. All signals were collected at a sampling rate of 200 points/s. The incubation time for each concentration of PQ was around 4.5 min.

## Whole-cell patch-clamp recording

A small aliquot of dissociated cells was placed in a 1-mL chamber mounted on the stage of an inverted microscope (Axio Observer Z1, Carl Zeiss). Cells were bathed at room temperature (25–27˚C) in HEPES-buffered Tyrode's solution. Ionic currents were recorded in the whole-cell configuration as previously described [21]. Patch electrodes were made from glass capillaries (o.d.: 1.5 mm, i.d.: 1.0 mm; A-M Systems, Sequim, WA, USA) using a two-stage vertical puller (P-830, Narishige, Tokyo, Japan) and were fire-polished. The electrode resistances were 2–5 MΩ when filled with normal pipette solution (containing in mM: KCl 120.0, NaCl 10.0, MgATP 5.0, EGTA 5.0, and HEPES 10.0, adjusted to pH 7.2 with KOH). Membrane currents were recorded using a voltage-clamp amplifier (Axopatch 200B, Molecular Devices, Sunnyvale, CA, USA). Electrode junction potentials (5–10 mV) were measured and nulled before cell suction. A high-resistance seal (5–10 GΩ) was obtained before membrane patch disruption. Usually, >5 min was allowed for adequate cell dialysis following membrane patch disruption and before initiating the voltage pulse protocol. Series resistances were compensated to minimize the duration of capacitive surge on the current recording and the voltage drop produced across the clamped cell membrane. About 60%–80% of series resistances were compensated. Cell capacitance was measured by calculating the capacitive transient's total charge movement in response to a 5-mV hyperpolarizing pulse. Command pulses were generated by a 12-bit digital-to-analog converter (Digidata 1320A, Molecular Devices) controlled by pCLAMP software (Molecular Devices, version 8).

During $K^+$ current measurements, $Ca^{2+}$ and $Na^+$ inward currents were blocked by adding 1 mM $Co^{2+}$ and 30 μM tetrodotoxin (TTX) to the bathing solution, respectively. To activate $K^+$ currents, cells were voltage-clamped at a holding potential of –80 mV, and currents (inward rectifier [$I_{K1}$] or transient outward [$I_{to}$] $K^+$ current) were elicited by 500-ms hyperpolarizing or depolarizing test pulses ranging from –140 to +60 mV. Steady-state inactivation of $I_{to}$ was examined with a double-pulse protocol: a conditioning 400-ms pulse to various potentials ranging from –80 to 0 mV was followed by a test depolarizing pulse to +60 mV. The holding potential was –80 mV. Each peak current was normalized to the maximum current measured and plotted as a function of the conditioning potential. The resultant curves were fitted by the Boltzmann equation to estimate half-inactivation potential ($V_h$) and slope factor ($k$). The twin-pulse protocol, which consisted of two identical 200-ms depolarizing pulses to +60 mV from a holding potential of –80 mV, was used to study the recovery of $I_{to}$ channels. Prepulse–test pulse intervals varied between 10 and 550 ms. The incubation time for each PQ concentration was around 4.5 min.

## Statistical methods and data analysis

Continuous data are presented as mean ± standard deviation (SD) unless otherwise indicated. Categorical data are expressed as frequency (in percentage). Statistical comparisons were made using SAS 9.1 (SAS Institute Inc., Cary, NC, USA), and $p$ values < 0.05 were considered statistically significant. The Student's $t$-test was used for univariate analysis of continuous variables. Generalized estimation equation (GEE) models were conducted to analyze the change of outcome variables over time and were employed in the analysis of variables in Table 1 and Fig 3.

**Table 1. Effects of PQ on hemodynamic and electrocardiographic variables in anesthetized rats.**

| Variables | Time (min) | | | | | | | | |
|---|---|---|---|---|---|---|---|---|---|
| | **0–20** | **30–50** | **60–80** | **90–110** | **120–140** | **150–170** | **180–200** | **210–230** | ***p* value** |
| **HR (beats/min)** | | | | | | | | | |
| Saline | 344 ± 45 | 332 ± 50 | 336 ± 45 | 336 ± 45 | 340 ± 47 | 342 ± 41 | 339 ± 43 | 343 ± 42 | – |
| 100 mg/kg PQ | 314 ± 36 | 303 ± 33 | 302 ± 37 | 302 ± 44 | 300 ± 50 | 286 ± 50 | 272 ± 44 | 276 ± 43 | <0.0001 |
| 180 mg/kg PQ | 344 ± 38 | 359 ± 50 | 366 ± 48 | 350 ± 62 | 348 ± 40 | 331 ± 31 | 287 ± 33 | 302 ± 9 | 0.001 |
| **SBP (mmHg)** | | | | | | | | | |
| Saline | 102.0 ± 20.5 | 92.8 ± 22.6 | 95.4 ± 24.9 | 91.5 ± 21.2 | 88.9 ± 25.2 | 84.0 ± 27.0 | 78.3 ± 33.1 | 80.6 ± 34.0 | – |
| 100 mg/kg PQ | 98.0 ± 22.7 | 95.2 ± 20.0 | 82.6 ± 30.5 | 73.6 ± 35.5 | 74.2 ± 33.4 | 61.9 ± 32.9 | 56.0 ± 37.7 | 60.1 ± 39.4 | 0.02 |
| 180 mg/kg PQ | 92.6 ± 25.6 | 97.8 ± 30.2 | 97.8 ± 31.3 | 82.7 ± 28.8 | 79.6 ± 25.6 | 68.1 ± 29.8 | 51.1 ± 13.9 | 47.5 ± 18.0 | 0.0001 |
| **DBP (mmHg)** | | | | | | | | | |
| Saline | 46.8 ± 10.5 | 41.1 ± 12.1 | 41.7 ± 13.8 | 38.7 ± 11.9 | 37.5 ± 14.5 | 36.4 ± 14.9 | 35.0 ± 17.2 | 36.2 ± 16.9 | – |
| 100 mg/kg PQ | 40.5 ± 10.9 | 34.8 ± 10.5 | 29.4 ± 12.9 | 26.4 ± 15.1 | 24.8 ± 13.6 | 21.0 ± 12.9 | 18.7 ± 14.6 | 20.7 ± 17.8 | 0.03 |
| 180 mg/kg PQ | 40.0 ± 10.8 | 40.3 ± 10.7 | 36.7 ± 10.7 | 29.1 ± 10.5 | 26.9 ± 9.7 | 21.9 ± 12.2 | 16.6 ± 3.9 | 14.7 ± 4.6 | 0.0001 |
| **MAP (mmHg)** | | | | | | | | | |
| Saline | 65.8 ± 12.2 | 59.4 ± 14.7 | 58.6 ± 15.8 | 56.6 ± 15.0 | 54.6 ± 16.9 | 52.9 ± 17.8 | 49.4 ± 20.8 | 50.5 ± 21.9 | – |
| 100 mg/kg PQ | 60.2 ± 14.1 | 54.2 ± 12.5 | 45.7 ± 17.0 | 41.0 ± 19.6 | 40.8 ± 19.2 | 36.1 ± 20.3 | 31.4 ± 21.8 | 34.1 ± 24.5 | 0.03 |
| 180 mg/kg PQ | 61.3 ± 15.5 | 58.8 ± 14.7 | 55.5 ± 16.1 | 50.0 ± 16.8 | 47.2 ± 14.8 | 38.6 ± 15.0 | 27.8 ± 6.4 | 27.6 ± 7.9 | 0.0002 |
| **LVESP (mmHg)** | | | | | | | | | |
| Saline | 105.1 ± 12.8 | 98.9 ± 14.5 | 98.4 ± 14.1 | 95.7 ± 14.3 | 92.6 ± 17.0 | 90.2 ± 18.3 | 85.6 ± 23.0 | 86.6 ± 23.3 | – |
| 100 mg/kg PQ | 96.6 ± 13.4 | 93.1 ± 12.8 | 84.8 ± 19.7 | 81.1 ± 24.0 | 81.2 ± 22.1 | 74.6 ± 25.4 | 67.7 ± 29.9 | 75.4 ± 24.5 | 0.11 |
| 180 mg/kg PQ | 103.9 ± 24.1 | 102.7 ± 26.5 | 101.8 ± 22.6 | 94.7 ± 26.8 | 95.1 ± 19.5 | 90.2 ± 17.5 | 77.8 ± 21.4 | 87.1 ± 30.3 | 0.0003 |
| **LVEDP (mmHg)** | | | | | | | | | |
| Saline | 4.6 ± 2.9 | 4.4 ± 2.9 | 4.4 ± 2.9 | 4.3 ± 3.1 | 4.0 ± 3.1 | 4.0 ± 3.2 | 3.7 ± 3.2 | 3.8 ± 3.5 | – |
| 100 mg/kg PQ | 1.4 ± 3.2 | 1.4 ± 3.1 | 1.7 ± 3.5 | 2.5 ± 3.4 | 3.4 ± 3.1 | 3.9 ± 3.2 | 3.8 ± 3.4 | 4.6 ± 4.2 | 0.01 |
| 180 mg/kg PQ | 0.4 ± 4.5 | 0.6 ± 4.4 | 1.6 ± 4.5 | 1.6 ± 4.3 | 1.6 ± 4.1 | 1.2 ± 3.0 | 0.6 ± 3.1 | 3.0 ± 2.0 | 0.75 |
| **+d$P$/d$t_{max}$ (mmHg/s)** | | | | | | | | | |
| Saline | 9406 ± 2171 | 8297 ± 2226 | 8336 ± 2225 | 8013 ± 2001 | 7695 ± 2351 | 7347 ± 2214 | 7039 ± 2848 | 7303 ± 2756 | – |
| 100 mg/kg PQ | 8853 ± 1311 | 8292 ± 1350 | 7341 ± 2267 | 6706 ± 3282 | 6779 ± 3059 | 5922 ± 2825 | 4684 ± 2819 | 4672 ± 2152 | 0.004 |
| 180 mg/kg PQ | 7602 ± 2931 | 7987 ± 3374 | 8482 ± 3522 | 7901 ± 3846 | 8041 ± 3725 | 7063 ± 3614 | 4909 ± 2721 | 6009 ± 2883 | 0.07 |
| **−d$P$/d$t_{max}$ (mmHg/s)** | | | | | | | | | |
| Saline | -4745 ± 1081 | -4240 ± 981 | -4245 ± 1116 | -3993 ± 931 | -3851 ± 1045 | -3706 ± 1081 | -3501 ± 1261 | -3547 ± 1129 | – |
| 100 mg/kg PQ | -4225 ± 1092 | -3867 ± 13134 | -3334 ± 1782 | -3350 ± 1889 | -3318 ± 1653 | -2677 ± 1577 | -2451 ± 1761 | -2778 ± 1591 | 0.05 |
| 180 mg/kg PQ | -4114 ± 2014 | -4509 ± 2434 | -4271 ± 2177 | -3608 ± 1951 | -3829 ± 2009 | -3426 ± 1851 | -2341 ± 650 | -2140 ± 714 | 0.01 |
| **P wave duration (ms)** | | | | | | | | | |
| Saline | 28.2 ± 5.8 | 28.3 ± 5.8 | 27.8 ± 6.2 | 28.7 ± 6.3 | 27.7 ± 6.1 | 27.9 ± 6.1 | 28.8 ± 5.9 | 32.2 ± 9.4 | – |
| 100 mg/kg PQ | 24.8 ± 3.4 | 25.1 ± 3.1 | 26.2 ± 4.5 | 27.6 ± 5.4 | 28.1 ± 6.5 | 26.8 ± 5.3 | 30.3 ± 8.6 | 33.1 ± 8.8 | 0.006 |
| 180 mg/kg PQ | 29.7 ± 7.8 | 30.5 ± 8.9 | 30.2 ± 8.4 | 33.2 ± 10.5 | 32.1 ± 10.8 | 30.9 ± 7.3 | 49.2 ± 13.5 | 47.9 ± 8.2 | 0.0008 |
| **PR (ms)** | | | | | | | | | |
| Saline | 60.3 ± 7.9 | 60.9 ± 7.8 | 60.5 ± 7.2 | 60.2 ± 8.3 | 60.2 ± 7.4 | 61.2 ± 7.2 | 61.0 ± 8.8 | 64.4 ± 11.5 | – |
| 100 mg/kg PQ | 66.9 ± 5.8 | 66.0 ± 4.9 | 67.0 ± 5.1 | 68.0 ± 6.7 | 68.8 ± 8.5 | 70.2 ± 8.3 | 78.7 ± 12.3 | 83.2 ± 13.4 | 0.02 |
| 180 mg/kg PQ | 60.9 ± 6.8 | 66.0 ± 21.4 | 60.8 ± 11.4 | 61.3 ± 14.2 | 62.5 ± 15.9 | 61.0 ± 11.4 | 79.2 ± 13.9 | 39.1 ± 36.2 | 0.047 |
| **QRS (ms)** | | | | | | | | | |
| Saline | 14.2 ± 1.0 | 14.1 ± 1.0 | 14.2 ± 0.9 | 13.9 ± 0.8 | 13.9 ± 0.9 | 13.8 ± 0.8 | 13.9 ± 1.0 | 14.5 ± 1.7 | – |
| 100 mg/kg PQ | 14.2 ± 1.3 | 13.7 ± 1.3 | 14.2 ± 1.4 | 15.9 ± 3.8 | 15.6 ± 3.4 | 16.4 ± 4.0 | 20.8 ± 7.8 | 22.4 ± 6.2 | 0.0002 |
| 180 mg/kg PQ | 14.0 ± 1.6 | 14.5 ± 3.6 | 14.1 ± 3.1 | 15.0 ± 5.3 | 13.3 ± 2.3 | 16.1 ± 3.0 | 24.7 ± 10.0 | 35.8 ± 29.0 | < .0001 |
| **QT (ms)** | | | | | | | | | |
| Saline | 65.1 ± 13.2 | 65.6 ± 11.3 | 66.8 ± 10.2 | 66.3 ± 10.7 | 66.7 ± 10.8 | 67.1 ± 9.9 | 68.1 ± 10.1 | 70.0 ± 9.9 | – |
| 100 mg/kg PQ | 77.1 ± 16.7 | 86.7 ± 11.5 | 86.4 ± 11.8 | 85.5 ± 14.0 | 92.1 ± 11.1 | 96.2 ± 14.0 | 96.5 ± 13.3 | 93.5 ± 21.0 | 0.02 |

*(Continued)*

**Table 1.** (Continued)

| Variables | Time (min) | | | | | | | | p value |
|---|---|---|---|---|---|---|---|---|---|
| | 0–20 | 30–50 | 60–80 | 90–110 | 120–140 | 150–170 | 180–200 | 210–230 | |
| 180 mg/kg PQ | 78.5 ± 6.8 | 83.5 ± 10.5 | 84.2 ± 15.0 | 86.8 ± 14.2 | 79.6 ± 6.5 | 83.5 ± 8.6 | 91.0 ± 11.3 | 101.4 ± 22.1 | < .0001 |
| QTc (ms) | | | | | | | | | |
| Saline | 66.2 ± 15.8 | 65.5 ± 13.6 | 67.1 ± 12.9 | 66.6 ± 12.6 | 66.5 ± 13.0 | 67.1 ± 12.0 | 68.1 ± 12.6 | 68.7 ± 11.6 | – |
| 100 mg/kg PQ | 74.3 ± 15.8 | 82.5 ± 9.8 | 82.1 ± 10.1 | 80.5 ± 14.3 | 85.8 ± 6.2 | 88.7 ± 7.1 | 85.5 ± 9.6 | 89.9 ± 13.8 | 0.61 |
| 180 mg/kg PQ | 79.6 ± 7.4 | 83.7 ± 10.9 | 84.8 ± 12.3 | 84.2 ± 12.3 | 77.4 ± 8.4 | 83.1 ± 7.3 | 84.5 ± 7.4 | 89.2 ± 10.9 | 0.04 |

Data are expressed as mean ± SD of $n = 10$ rats per group. $p$ value of slope with time effect compared to saline group. HR, heart rate; SBP, systolic blood pressure; DBP, diastolic blood pressure; MAP, mean arterial blood pressure; LVESP, left ventricular end systolic pressure; LVEDP, left ventricular end diastolic left ventricle pressure; +d$P$/d$t_{max}$ and –d$P$/d$t_{max}$, maximal rate of ringing and fall of LV pressure, respectively; QTc, rate-corrected QT interval derived using normalized Bazett's formula QTc = QT/(RR/$f$)$^{1/2}$, where $f$ = 180 ms.

The body weight of rats was also adjusted in fitting GEE models. The outcome variable distribution was proved to be Gaussian by a normality test. An exchangeable working correlation matrix was applied when applying the GEE method. An equation of the form fitted concentration-response curves: $E = E_{max}/[1+(IC_{50})^{nH}]$, where $E$ is the effect at concentration C, $E_{max}$ is the maximal effect, $IC_{50}$ is the concentration for half-maximal inhibition and $n_H$ is the Hill coefficient. The conductance of $I_{to}$ ($G_{to}$) was calculated according to the equation $G_{to} = I_{to}/(V_m-V_{rev})$, where $V_{rev}$ is the reversal potential of $I_{to}$. The activation curves of $I_{to}$ were fitted by the Boltzmann equation: $G_{to}/G_{to, max} = 1/\{1+\exp[(V_h-V_m)/k]\}$, where $G_{to, max}$ is the maximal ionic conductance, $V_h$ is the half-maximal activation potential, $V_m$ is the membrane potential, and $k$ is the slope factor. The inactivation curves of $I_{to}$ were fitted by the Boltzmann equation: $I_{to}/I_{to, max} = 1/\{1+\exp[(V_m-V_h)/k]\}$; where $I_{to}$ gives the current amplitude and $I_{to, max}$ is its maximum, $V_m$ is the prepulse potential, $V_h$ is the half-maximal inactivation potential, and $k$ is the slope factor.

## Results

### Effects of PQ on hemodynamic measurements and ECG

Baseline hemodynamic and ECG parameters did not significantly vary among rats receiving the saline vehicle or PQ at a dose of 100 or 180 mg/kg (S1 Table). Fig 1 shows a representative example of the PQ effect (180 mg/kg) on arterial pressure, LV pressure, first derivative of LV pressure (LV d$P$/d$t$), and the ECG at different time points. PQ impeded LV performance in both systolic and diastolic phases, decreased blood pressure, and slowed heart rates. PQ also prolonged P wave duration and the PR, QRS, QT, and QTc intervals in a dose-dependent manner (Table 1). In contrast, the infusion of the corresponding volume of vehicle (normal saline) produced no significant changes in any of the parameters (S1 Fig and Table 1). All the animals treated with saline vehicles survived until the end of the experiment. The calculated mortality rate of 100 and 180 mg/kg PQ-treated rats was 60% (6/10; $p < 0.05$) and 90% (9/10; $p < 0.001$), respectively. The higher mortality rate was closely correlated with a longer QTc interval of rats following PQ infusion.

### Effects of PQ on contractile force in Langendorff-perfused rat hearts

The acute effects of PQ (33 or 60 μM) on contractility in LV myocardium were examined in Langendorff-perfused rat hearts. As shown in Fig 2, PQ decreased LV developed pressure (LVDP) and the maximal rate of rising (+d$P$/d$t_{max}$) and fall (–d$P$/d$t_{max}$) of LV pressure in

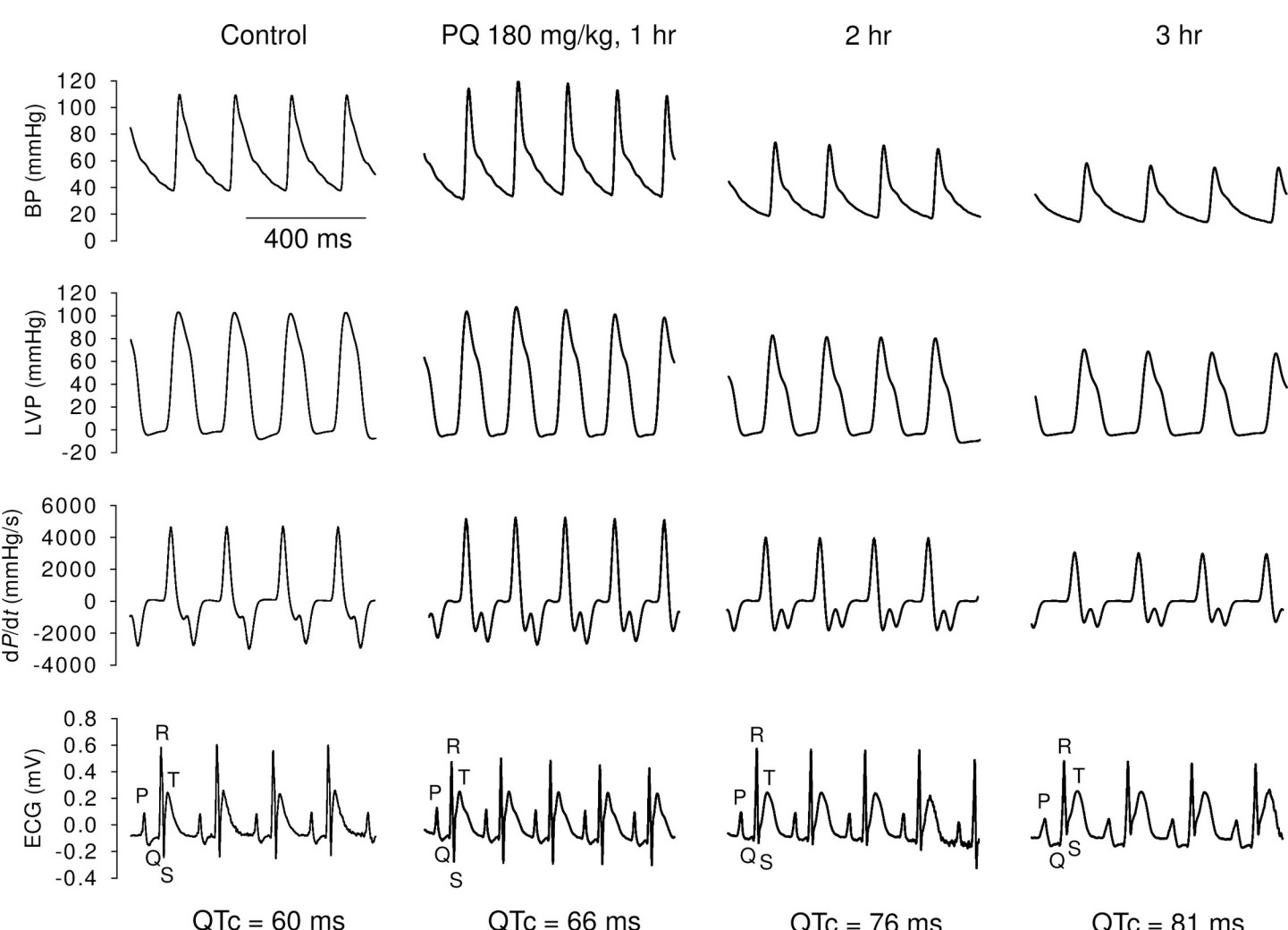

**Fig 1. Representative recordings of arterial pressure, LV pressure (LVP), first derivative of LV pressure (LV d$P$/d$t$), and ECG from an anesthetized rat at baseline and at various times after PQ treatment (180 mg/kg, i.v.).** QTc value in each panel denotes rate-corrected QT interval derived using normalized Bazett's formula QTc = QT/(RR/$f$)$^{1/2}$, where $f$ = 180 ms.

isolated hearts in a concentration-dependent manner. On average ($n$ = 10 for both groups), LVDP decreased from a baseline value of 30.1 ± 10.7 to 18.1 ± 6.9 mmHg and from 25.6 ± 6.4 to 8.5 ± 5.1 mmHg after 60 min treatment with 33 and 60 μM PQ, respectively ($p < 0.0001$). LV +d$P$/d$t_{max}$ decreased from a baseline value of 950.7 ± 286.3 to 581.6 ± 183.7 mmHg/s and from 773.3 ± 164.8 to 272.4 ± 167.7 mmHg/s with the application of 33 and 60 μM PQ, respectively ($p < 0.0001$). LV–d$P$/d$t_{max}$ decreased from –600.8 ± 271.4 to –387.1 ± 167.5 mmHg/s and from –532.1 ± 116.4 to –177.0 ± 105.4 mmHg/s with the application of 33 and 60 μM PQ, respectively ($p < 0.0001$). In contrast, administration of the corresponding volume of normal saline had no changes in any of the LV pressure parameters (S2 Fig).

## Effects of PQ on intracellular Ca$^{2+}$ transients and cell shortening in rat ventricular myocytes

Fig 3 shows the continuous (panel A) and expanded recordings (panel B) and the summarized data (panels C and D) of the PQ effects (10–60 μM) on Ca$^{2+}$ transients (fura-2 fluorescence

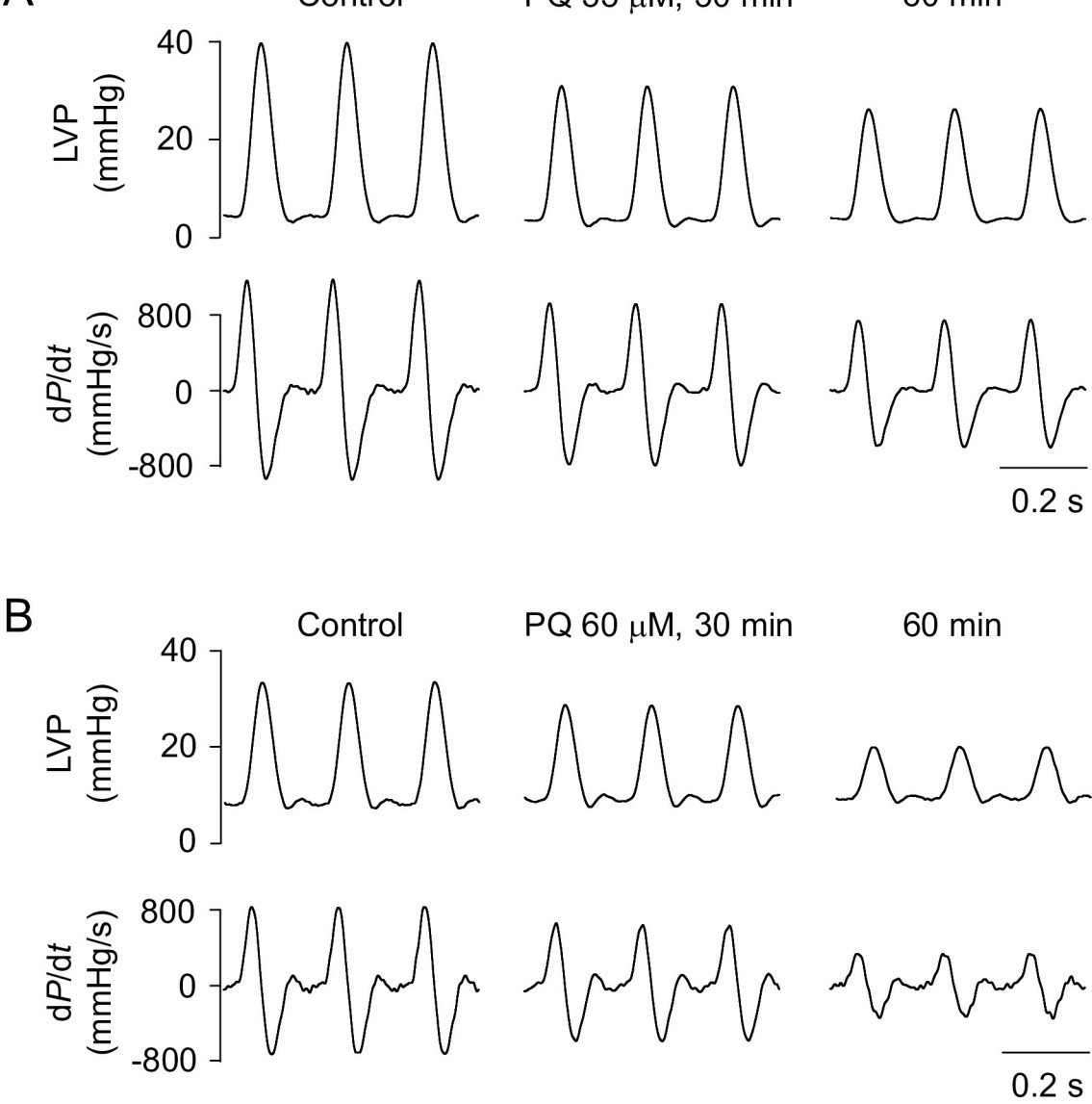

**Fig 2.** Representative tracings of LVP (upper) and d$P$/d$t$ (lower) signals recorded from Langendorff-perfused rat hearts paced at 300 beats/min at baseline and following treatment with 33 μM (A) or 60 μM PQ (B).

ratio $F_{340}/F_{380}$) and cell shortening in rat ventricular myocytes. PQ decreased cell shortening in a concentration-dependent manner (Fig 3C). However, PQ did not affect the time-to-peak of cell shortening and time to 50% relengthening (S3A and S3B Fig). PQ also decreased the amplitude of the fluorescence ratio (Ca$^{2+}$ transients) in a concentration-dependent manner (Fig 3D). However, it had no apparent effect on the time to peak (S3C Fig) and decay time constant of the Ca$^{2+}$ transients (S3D Fig). However, the application of the corresponding volumes of normal saline produced no changes in any of these parameters (S4 Fig).

### Effects of PQ on K$^+$ currents in rat isolated ventricular myocytes

To separate the K$^+$ currents from overlapping currents, Na$^+$ and Ca$^{2+}$ inward currents were blocked with 30 μM TTX and 1 mM Co$^{2+}$, respectively. Typical current traces recorded in

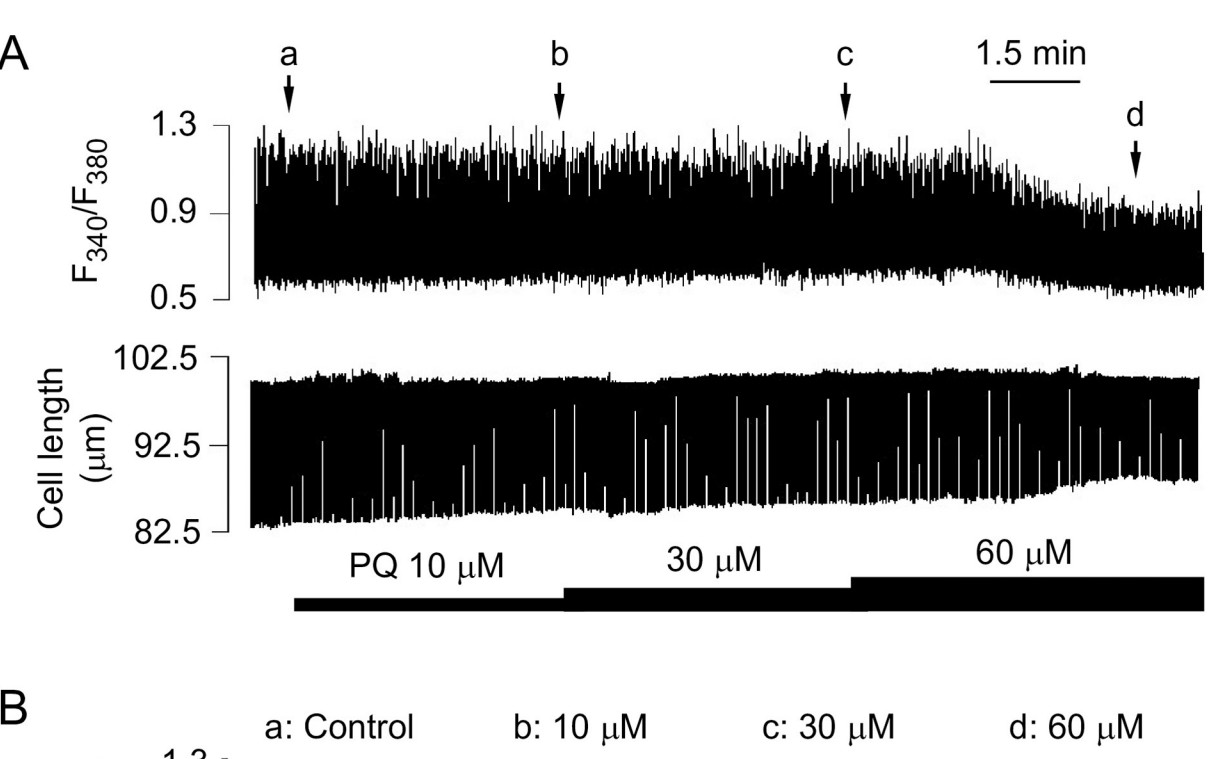

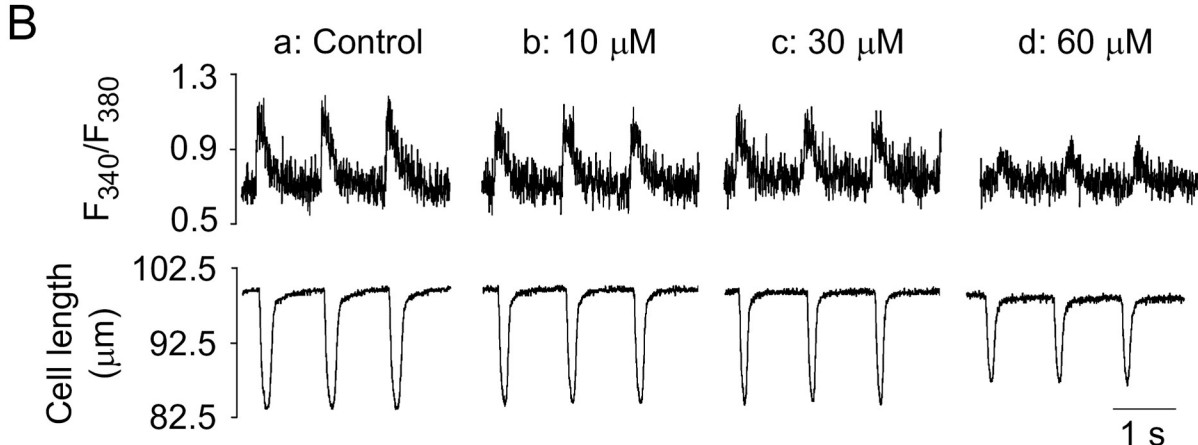

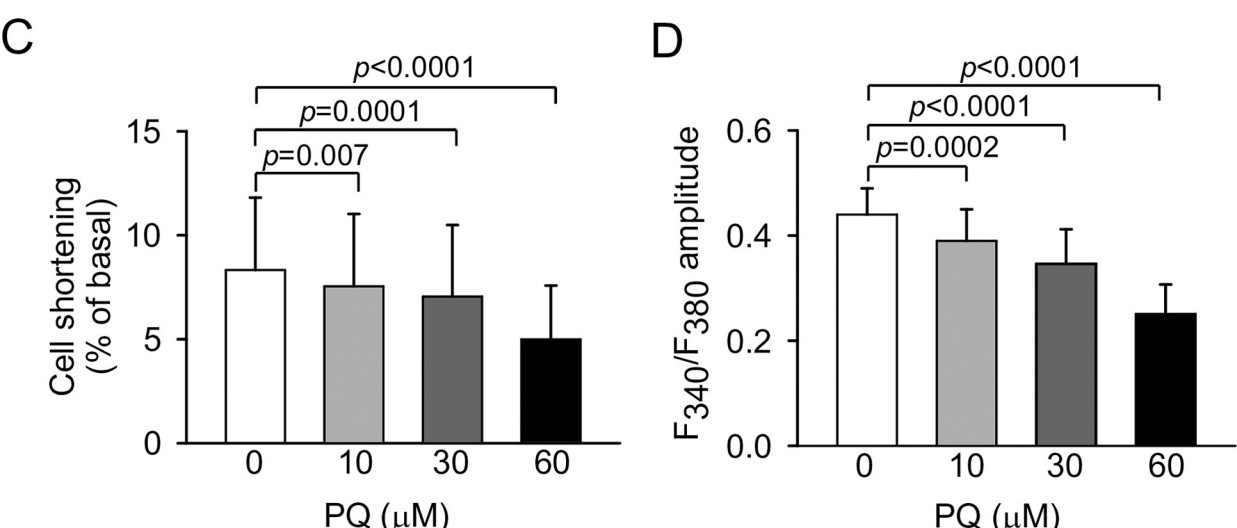

**Fig 3. Effects of PQ on Ca$^{2+}$ transients (represented by fura-2 fluorescence ratio $F_{340}/F_{380}$) and cell shortening in rat ventricular myocytes paced at 1 Hz.** (A) Continuous recordings of Ca$^{2+}$ transients (upper panel) and cell shortening (lower panel) showing the effects of the cumulative application of 10, 30, and 60 μM PQ. (B) Recordings on an expanded time scale taken at the time indicated by the corresponding letters over the $F_{340}/F_{380}$ trace in A. (C, D) The mean data of the amplitude of cell shortening (C) and Ca$^{2+}$ transient (D) before and after application of PQ. Data are expressed as mean ± SD ($n = 11$). Cell shortening was normalized to resting cell length.

response to depolarizing and hyperpolarizing steps to test potentials between +60 and −140 mV from a holding potential of −80 mV are shown in Fig 4A. PQ (3 and 10 μM) moderately reduced the amplitude of peak outward K$^+$ current ($I_{to}$) but had no significant effect on inward rectifier K$^+$ current ($I_{K1}$). PQ also reduced the amplitude of steady-state outward K$^+$ current ($I_{ss}$) at the end of 400-ms long clamp steps. Fig 4B and 4C shows the current density–voltage ($I$–$V$) relationship for the peak currents and $I_{ss}$ before and after the addition of PQ (1, 3, and 10 μM), respectively. The percent inhibition of $I_{to}$ integral by PQ was not significantly dependent on the step potentials (Fig 4D).

The effect of PQ on $I_{to}$ was investigated further by analyzing its concentration dependence. $I_{to}$ was elicited by a depolarizing pulse to +40 mV from a holding potential of −80 mV. Fig 4E shows superimposed K$^+$ current traces before and after cumulative superfusion with 3 and 10 μM PQ. The decay of currents during an activating clamp step in control conditions and after administration with 1, 3, and 10 μM PQ were well fitted by a single exponential function: $I_{to}$ (t) = A$_1$ exp (−t/τ) + A$_0$, where A$_1$ and τ are the initial amplitude and time constant of inactivation, respectively. A$_0$ is a time-independent component. The average value of the decay time constant (τ) was 38.7 ± 13.4 ms ($n = 5$) under control conditions. In the presence of 1, 3, and 10 μM PQ, decay τ were 37.7 ± 1.3, 49.3 ± 16.2, and 52.1 ± 19.8 ms, respectively ($p = 0.26$, $n = 5$). PQ had no significant effect on the decay time course of $I_{to}$. Fig 4F illustrates the percent reduction of the $I_{to}$ integral as a function of the PQ concentration logarithm. The data were fitted with a Hill equation to obtain a concentration-response curve. The calculated $IC_{50}$ for $I_{to}$ was 2.4 μM, with an $E_{max}$ of 35.4% inhibition and an $n_H$ of 2.4 ($n = 5$).

## Effects of PQ on steady-state activation, inactivation, and recovery from inactivation of $I_{to}$

The predrug superimposed current traces are shown in Fig 5A, and the voltage dependence of steady-state activation and inactivation curves of $I_{to}$ are shown in Fig 5B. PQ caused a leftward shift of the steady-state inactivation relationship of $I_{to}$ without affecting its slope factor $k$ (Fig 5B). Under control conditions ($n = 5$), mean half-inactivation potential ($V_h$) was calculated as −24.7 ± 9.8 mV and $k$ as −4.9 ± 1.9 mV. Mean $V_h$ were −39.9 ± 5.6, −48.5 ± 6.6, and −50.5 ± 8.1 mV and $k$ were −4.9 ± 2.0, −4.9 ± 2.0, and −4.0 ± 0.3 mV in the presence of 1, 3, and 10 μM PQ, respectively ($p = 0.007$ for $V_h$ and $p = 0.7$ for $k$). The activation curves shown in Fig 5B were obtained from the normalized conductance of $I_{to}$ channels ($G_{to}/G_{to, max}$) calculated from the $I_{to}$ amplitude data obtained in Fig 4A and 4B. Mean half-activation potential ($V_h$) and $k$ values were 14.9 ± 11.9 and 10.4 ± 1.6 mV ($n = 4$), respectively, under control conditions. PQ caused a leftward shift of the voltage dependence for activation to negative potentials. Mean $V_h$ were −2.5 ± 7.7, −14.4 ± 15.2, and −9.6 ± 9.1 mV and $k$ were 11.3 ± 0.9, 18.9 ± 3.9, and 21.4 ± 4.4 mV in the presence of 1, 3, and 10 μM PQ, respectively ($p = 0.029$ for $V_h$ and $p = 0.012$ for $k$). The effect of PQ on the recovery kinetics of $I_{to}$ was also examined (Fig 5C and 5D). Recovery curves in the absence and presence of PQ were well fitted by a single-exponential function. As shown in Fig 5D, PQ had no significant effect on the recovery time course of $I_{to}$. The mean recovery time constant was 44.2 ± 31.5 ms ($n = 5$) under control conditions. Recovery time constants of $I_{to}$ were 51.8 ± 33.0, 64.6 ± 45.1, and 94.2 ± 108.3 ms ($p = 0.38$) in the presence of 1, 3, and 10 μM PQ, respectively.

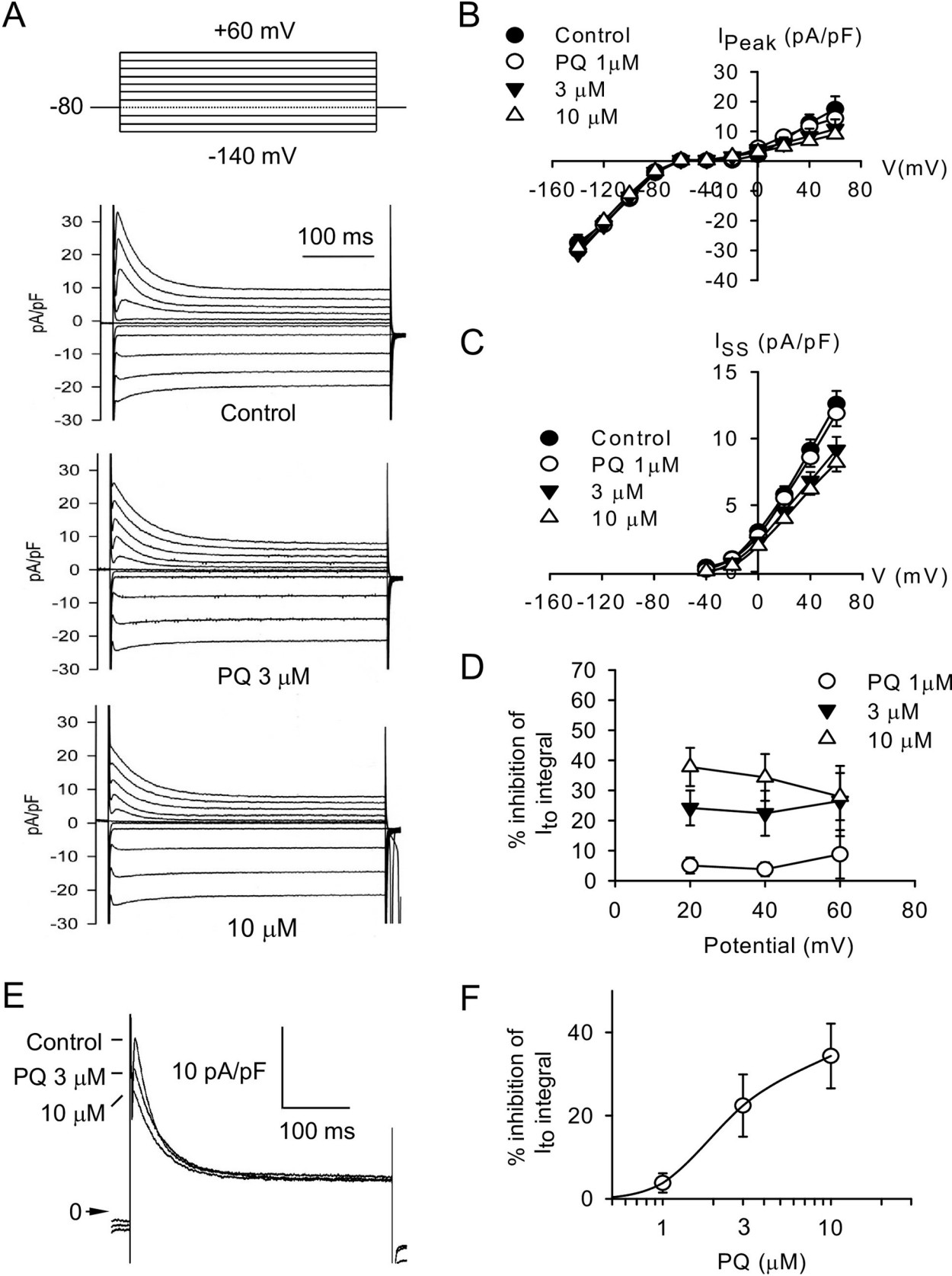

**Fig 4. Effect of PQ on K$^+$ currents.** (A) Families of current traces elicited by a series of 400-ms long depolarizing or hyperpolarizing pulses from a holding potential of −80 mV in the absence and presence of 3 and 10 μM PQ. (B, C) Averaged *I–V* relationship for $I_{to}$ and $I_{K1}$ peak currents (B) and $I_{ss}$ (C) observed in the absence, and presence of 1, 3, and 10 μM PQ. Peak $I_{to}$ current was measured as the difference between the peak current and the steady-state current at the end of the pulse. $I_{ss}$ current was measured as the steady-state current at the end of the pulse. Each data point indicates mean ± SD from 5 myocytes. (D) Percent inhibition of the $I_{to}$ integral by 1, 3, and 10 μM PQ calculated at different depolarizing potentials. Data points are mean ± SD (*n* = 5). (E) Original **s**uperimposed families of current traces generated by 400-ms depolarizing pulses to +40 mV from a holding potential of −80 mV in the absence or presence of increasing PQ concentrations. The *arrow* indicates the zero current density level. (F) Concentration–response curve for the effect of PQ on the integral of $I_{to}$ at +40 mV. Data points are mean ± SD (*n* = 5). The *continuous line* was drawn according to the fitting of the Hill equation.

## Discussion

The present study investigated the hemodynamic and cardiac electromechanical effects of acute paraquat exposure in rats. The major findings of this study are as follows: (1) PQ decreased heart rate, arterial blood pressure, and cardiac contractility, as well as prolonged the PR, QRS, QT, and QTc intervals in a dose-dependent manner in anesthetized rats; (2) PQ decreased LV developed pressure and contractility in the Langendorff-perfused rat hearts; (3) PQ decreased both the amplitude of Ca$^{2+}$ transients and fractional cell shortening in a concentration-dependent manner in rat ventricular myocytes; and (4) PQ suppressed $I_{to}$ and $I_{ss}$ channels and reduced the availability and altered the gating kinetics of $I_{to}$ channels. The findings provide novel evidence for acute PQ exposure-induced cardiac performance and mechanical dysfunctions as well as electrical disturbances.

A broad spectrum of cardiovascular effects ranging from minimal changes on the ECG to acute and extensive myocardial necrosis has been clinically observed in human acute PQ poisoning [10]. Severely poisoned patients expire from a rapid progression of myocardial depression and irreversible circulatory shock in the acute and subacute phases of PQ poisoning [22]. A rapid PQ accumulation into the heart but not in the lung or kidney is a significant cause of mortality in the early stage of PQ poisoning when a large amount of paraquat (364 mg/kg) was ingested in rats [7]. Previous studies have reported that the PQ dose is a critical prognostic factor for predicting mortality in PQ poisoning patients [23,24]. The clinical presentation of PQ intoxication can be distinguished by the following groups: (1) mild poisoning (<20 mg/kg BW) where the patients have minor symptoms in the gastrointestinal system; (2) moderate to severe poisoning (20–40 mg/kg BW) where the patients develop renal and lung injuries and mortality is usually delayed for 2–3 weeks; and (3) fulminant poisoning (>40 mg/kg BW) wherein this PQ dose causes multiple organ failure, leading to mortality within hours and never delayed for more than a few days [7]. The lethal doses of PQ poisoning in rodents are generally much higher compared with those in humans [7]. Here, the dosages of 100 and 180 mg/kg (approximately twice and triple LD$_{50}$ in a rat study, respectively) were used for *in vivo* study and a concentration range of 1–60 μM *in vitro* to simulate the mild, moderate to severe, and massive poisoning in clinical situations. Although the intoxication dosages are much higher in rats, the current study showed that the dose-dependent cardiac suppressive effects of PQ are comparable to the clinical conditions in PQ poisoning cases.

Our *in vivo* study revealed that PQ exposure exhibited a dose- and time-dependent hypotensive effect. A previous study showed that PQ did not cause any vasorelaxant effect on phenylephrine-precontracted rat aortic rings with intact endothelium [25]. It is conceivable that PQ-induced hypotensive response may be attributable mainly to its direct cardiac suppressive effects as manifested by its reduction of heart rate and LV contractility. In general, any decreases in either the depolarizing (e.g., funny channel, T-type or L-type Ca$^{2+}$ currents) or the repolarizing (e.g., delayed outward K$^+$ current) currents of the SA node may all cause the bradycardiac effect [26]. Thus, further studies are needed to clarify whether PQ could suppress the SA nodal ionic currents and thereby decrease the heart rate. PQ exposure prolonged the

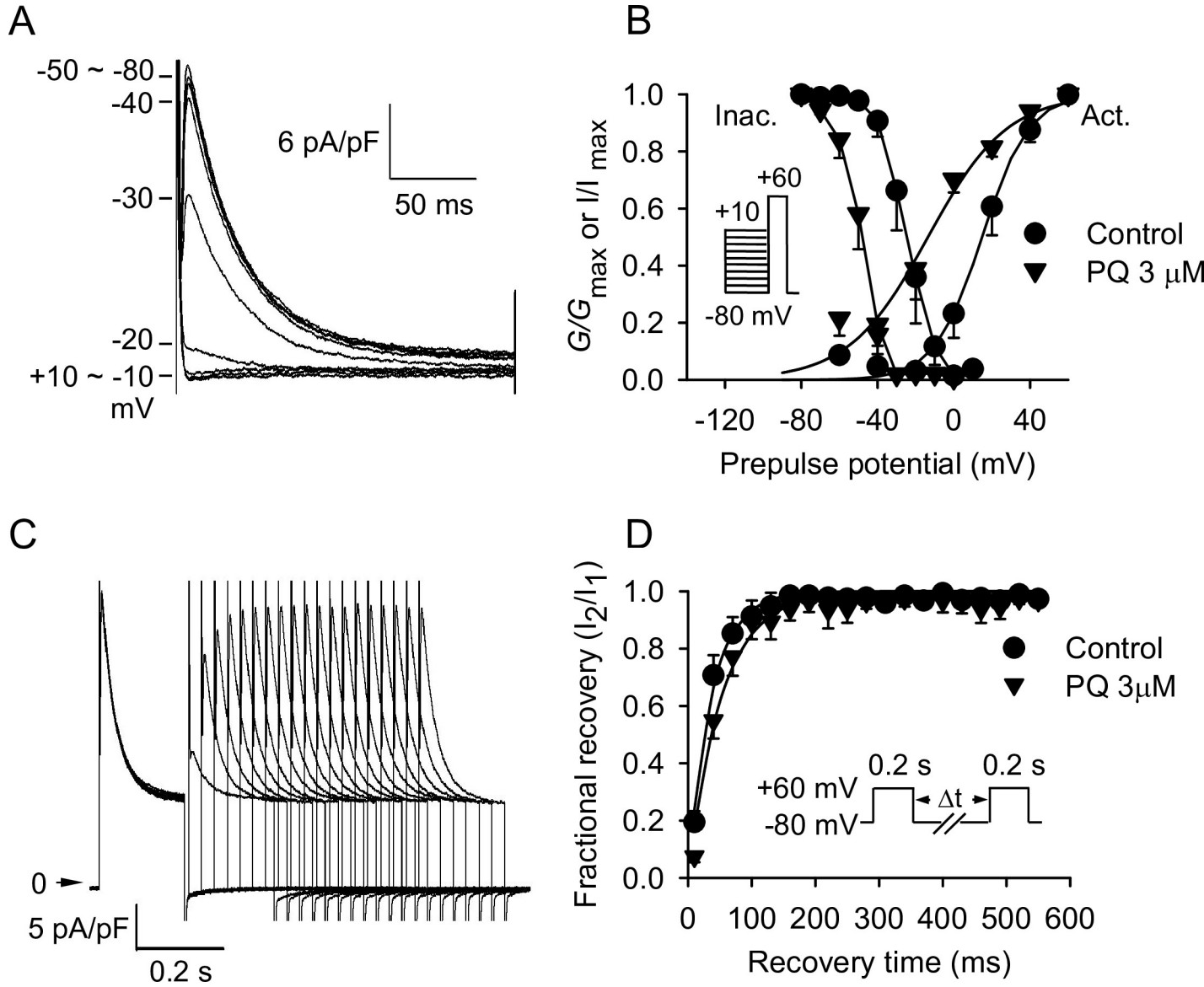

**Fig 5. Voltage dependence of steady-state $I_{to}$ activation and inactivation in the absence and presence of 3 μM PQ.** Steady-state inactivation was examined with a double-pulse protocol: A conditioning 400 ms pulse to various potentials ranging from −80 to +10 mV was followed by a test depolarizing pulse to +60 mV (B, inset). The holding potential was −80 mV. The predrug superimposed current traces are shown in A. The inactivation curves for $I_{to}$ were obtained by normalizing the current amplitudes ($I$) to the maximal value ($I_{max}$) and plotted as a function of the conditioning potentials before and after PQ ($n$ = 5). The activation curves were obtained from the normalized conductance of $I_{to}$ channels ($G_{to}/G_{to, max}$), which were calculated from the $I_{to}$ amplitude in Fig 4B and plotted as a function of the depolarizing potentials ($n$ = 5). The *solid lines* drawn through the data points were best fitted to the Boltzmann equation. (C, D) Effects of PQ on reactivation of $I_{to}$. The twin-pulse protocol consisted of two identical 200 ms depolarizing pulses to +60 mV from a holding potential of −80 mV (D, inset), and the prepulse–test pulse interval varied between 10 and 550 ms. An example of the recovery of $I_{to}$ from inactivation in control conditions is shown in C. The normalized currents (fractional recovery) obtained in the absence and presence of 3 μM PQ were plotted as a function of the recovery time. The *solid lines* represent a single exponential fit to the data in the absence and presence of 3 μM PQ ($n$ = 5), respectively.

QRS duration suggesting that the ventricular excitability could also be suppressed. As PQ administration also dose-dependently prolonged the P wave duration and P-R interval, which denote atrial depolarization time and atrial-ventricular (AV nodal) conduction interval, respectively, some mortality rate arising from severe SA nodal/atrial or AV nodal blockade is possible.

During cardiac excitation-contraction coupling, $Ca^{2+}$ enters the myocyte mainly through the L-type $Ca^{2+}$ channels following an action potential and triggers further $Ca^{2+}$ release from the sarcoplasmic reticulum (SR) [26]. The PQ-induced negative inotropic effect in the present study could be explained by its reduction of the amplitude of intracellular $Ca^{2+}$ transients. Depression of contractility was also observed in ventricular myocytes obtained from rats chronically treated with a lower PQ dose (10 mg/kg/day) for 3 weeks [27]. Although the authors observed a reduction in the amplitude of cell shortening, any notable change in the amplitude and kinetics of $Ca^{2+}$ transients was not observed. The authors speculated that decreased cell shortening could be explained by altered myofilament sensitivity to $Ca^{2+}$ when using lower PQ doses at chronic exposure. However, decreased cell shortening could be attributed to the reduced amplitude of $Ca^{2+}$ transients during acute- and high-dose PQ exposure as shown in the present study. In addition to changes in $Ca^{2+}$ transients, PQ-induced ROS production and subsequent development of oxidative stress may cause defects in transporter and channel functions [28], leading to perturbed $Ca^{2+}$ homeostasis and cardiac dysfunction. A previous study has demonstrated that PQ exposure-induced myocardial damage and contractile dysfunction are mediated by TLR4 activation and the associated secretion of proinflammatory cytokines [14]. Another study revealed that the endothelin system may be involved in the cardiac dysfunction triggered by PQ exposure for 48 h in mice, as evidenced in endothelin receptor A knockout attenuating the PQ-induced contractile dysfunction and $Ca^{2+}$ mishandling [15]. More recently, a study conducted by Zhang *et al.* suggested that the RIP1/RIP3/MLKL-dependent necroptosis pathway with oxidative stress may also contribute to the deranged $Ca^{2+}$ handling and contractile dysfunction in mice treated with PQ (45 mg/kg) for 48 h [29]. In contrast to the aforementioned pieces of literature [15,29], which showed that the smaller amplitude of $Ca^{2+}$ transient was associated with a longer decay time course in cardiomyocytes from animals with PQ exposure, our *in vitro* study demonstrated a similar effect in $Ca^{2+}$ transient amplitude but with no change in the decay time in myocytes following short-term PQ incubation. The differential effects can be attributable to the difference in experimental conditions (e.g., animal species, exposure dose/time course, *ex vivo* or *in vitro*, and so on). The longer decay time of $Ca^{2+}$ transients in previous *ex vivo* studies may implicate that the sequestration of $Ca^{2+}$ by SR $Ca^{2+}$ pump is impaired and may thereby decrease the $Ca^{2+}$ content of SR [25]. In our study, the PQ-induced decrease in $Ca^{2+}$ transients could be possibly due to its suppression of $Ca^{2+}$ influx through the $Ca^{2+}$ channels because no delay of decay time was observed. Further studies are needed to clarify this issue.

Until now, the underlying mechanism of PQ-induced cardiac electrical defects is not yet defined. This study provided an important new insight into the cellular mechanism underlying the PQ-induced QTc prolongation. Voltage-gated $K^+$ channels are known to play a crucial role in determining the shape and duration of the cardiac action potential. $I_{to}$ is generally considered an important repolarizing current in the mammalian action potential, including in human and rat atrium and ventricle [30,31]. The data showed that $I_{to}$, which is responsible for the early phase of repolarization, and $I_{ss}$, which may contribute to the late phase of repolarization of the rat cardiac action potential, were differentially blocked by PQ exposure. PQ did not affect the $I_{to}$ decay time suggesting that it did not affect the conversion of open channels to an inactivated state. Furthermore, the kinetic analysis showed that PQ caused a leftward shift of the $I_{to}$ inactivation curve and affected the voltage dependence for activation while it did not affect its recovery from inactivation. This finding suggests that PQ may preferentially bind inactivated channels, which could thereby decrease the number of resting $I_{to}$ channels available for activation. The suppression of $K^+$ currents may retard ventricular repolarization and contribute to the prolongation of action potential duration and QTc interval which may promote the occurrence of triggered activity and arrhythmias [12].

The prolonged QTc interval has been clinically associated with mortality from intoxication with several kinds of pesticides (e.g., PQ and organophosphates) [11,32]. QTc prolongation also affects mortality rates in patients with a variety of cardiac diseases (e.g., coronary artery disease and congestive heart failure) [33,34]. Moreover, prolonged QTc interval seems to correlate with poorer LV function. For example, QTc interval may represent an additional marker of LV systolic dysfunction in patients with anterior acute myocardial infarction [35]. Moreover, prolonged QTc interval appeared to correlate with LV dysfunction observed by echocardiography in patients who received anthracycline treatment [36]. In this context, the observations in this study such as prolongation of the QTc interval and suppression of contractility in isolated hearts, together with decreased cell shortening in ventricular myocytes may contribute to the cardiotoxicity of acute PQ poisoning.

Treatment of the most severely PQ-intoxicated patients remains a tremendous challenge nowadays. One of the current treatment modalities is immunosuppressive therapy. Some authors claimed that immunosuppressive therapy may have benefits in treating PQ poisoning patients who expired from lung fibrosis-related hypoxemia [37,38]. However, the methodological problems limit its application [39]. Moreover, the largest randomized controlled trial completed to date also reported no benefit of this therapy [40]. It seems that immunosuppressive therapy should have no beneficial effect in treating the most severe form of PQ intoxicated patients who suffered from a cardiovascular collapse in the early stage of poisoning. Other therapeutic strategies seem promising in animal studies mostly emerging from the suppression of PQ-induced oxidative stress or inflammatory response. Among them, lysine acetylsalicylate (200 mg/kg), which also can chelate PQ, confers a potent protective effect against PQ toxicity [41]. Atorvastatin, a lipid lowering drug, has been recently shown to attenuate PQ-induced cardiotoxicity in a rat model through similar mechanisms [42]. The results from a retrospective study demonstrated that edaravone, a free radical scavenger with an anti-inflammatory effect, protected the kidneys and liver but did not reduce pulmonary fibrosis and survival rate in patients with paraquat poisoning [43]. Currently, finding an effective strategy for the treatment of PQ-induced toxicity is attracting the attention of researchers.

## Conclusions

In summary, this study demonstrated that acute paraquat exposure produces negative hemodynamic and electromechanical effects in rat hearts. The cellular-suppressive effect on the contractility and voltage-gated $K^+$ channels of the myocytes could be applied with caution to explain the inevitable death of fulminant paraquat poisoning cases. It seems no efficient treatment modalities may be currently used to reverse the harmful effects on the heart in paraquat poisoning patients. However, an improved understanding of how PQ causes cardiac suppression and electrical disturbances provided by the current study may aid in the development of new treatment modalities.

## Supporting information

**S1 Fig. Representative recordings of arterial pressure, LV pressure (LVP), first derivative of LV pressure (LV d*P*/d*t*), and ECG from an anesthetized rat at baseline and at various times after treatment with normal saline (1 mL/kg, i.v.).**
(DOCX)

**S2 Fig. Effects of saline vehicle (0.03%) on LV pressure (LVP) in isolated hearts paced at 300 beats/min.**
(DOCX)

**S3 Fig. Effects of paraquat on kinetic parameters of cell shortening and intracellular Ca$^{2+}$ transients in rat ventricular myocytes.**
(DOCX)

**S4 Fig. Effects of saline vehicle on Ca$^{2+}$ transients (represented by fura-2 fluorescence ratio $F_{340}/F_{380}$) and cell shortening in rat ventricular myocytes paced at 1 Hz.**
(DOCX)

**S1 Table. Baseline hemodynamic and electrocardiographic parameters in anesthetized rats receiving the vehicle or PQ at a dose of 100 or 180 mg/kg.**
(DOCX)

## Author Contributions

**Conceptualization:** Chih-Chuan Lin, Gwo-Jyh Chang.

**Data curation:** Gwo-Jyh Chang.

**Formal analysis:** Kuang-Hung Hsu, Chia-Pang Shih, Gwo-Jyh Chang.

**Funding acquisition:** Chih-Chuan Lin.

**Methodology:** Chih-Chuan Lin, Gwo-Jyh Chang.

**Project administration:** Gwo-Jyh Chang.

**Supervision:** Gwo-Jyh Chang.

**Validation:** Chih-Chuan Lin, Gwo-Jyh Chang.

**Writing – original draft:** Chih-Chuan Lin.

**Writing – review & editing:** Chih-Chuan Lin.

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
