## [Decision Letter · Decision Letter 0]

10 Sep 2020

PONE-D-20-15928

Hemodynamic and electromechanical effects of paraquat in rat heart

PLOS ONE

Dear Dr. Chang,

Thank you for submitting your manuscript to PLOS ONE. After careful consideration, we feel that it has merit but does not fully meet PLOS ONE’s publication criteria as it currently stands. Therefore, we invite you to submit a revised version of the manuscript that addresses the points raised during the review process.

Thank you for submitting "Hemodynamic and electromechanical effects of paraquat in rat heart". Two reviewers have completed review and have raised some valid critiques which should be addressed upon revison. In addition, I would like you to address reasons for the relatively low hemodynamic values observed in the langendorff perfused hearts ( perfusion pressure etc...). Also relating to the langendorff experiments, it is not clear if separate untreated control hearts were run for the duration (60 min) or if the shown control is just pre-values, an untreated 60 min control would be ideal. Please clarify this in addition to the reviewers comments in your response. There were some noted strengths of the work identified including the patch clamp analysis and other figures, and we would like to reconsider a revised version of the manuscript.

Please submit your revised manuscript within 3-4 months. If you will need more time than this to complete your revisions, please reply to this message or contact the journal office at plosone@plos.org. Please include the following items when submitting your revised manuscript:

We look forward to receiving your revised manuscript.

Kind regards,

Richard T. Clements, PhD

Academic Editor

PLOS ONE

Journal Requirements:

2.  Please clarify how many rats were used in the in vivo experiment, and how many were in the control group versus the paraquat group.

3. Please clarify how many rats were used in the in vitro experiment.

4. Please carefully proofread your manuscript for typographical errors. For example, “The rats was anesthetized with pentobarbital sodium …” should be the “The rats were anesthetized with pentobarbital sodium …”

5. Please specify in your methods section the method of sacrifice, particularly for the control rats in the in vivo experiment.

6. Please ensure that you refer to Figure 6 in your text as, if accepted, production will need this reference to link the reader to the figure.

7. Please include a caption for figure 6.

Reviewers' comments:

Reviewer's Responses to Questions

**Comments to the Author**

1. Is the manuscript technically sound, and do the data support the conclusions?

Reviewer #1: Yes

Reviewer #2: Yes

2. Has the statistical analysis been performed appropriately and rigorously? 

Reviewer #1: No

Reviewer #2: Yes

3. Have the authors made all data underlying the findings in their manuscript fully available?

Reviewer #1: Yes

Reviewer #2: Yes

4. Is the manuscript presented in an intelligible fashion and written in standard English?

Reviewer #1: No

Reviewer #2: Yes

5. Review Comments to the Author

Reviewer #1: Chang et al. investigated the cardiotoxic effect of acute paraquat poisoning in a rat model. The study focused on contractile performance and electrophysiology of rat hearts and isolated cardiac myocytes using high doses of paraquat. The authors show an inhibitory effect of paraquat on heart rate, cardiac contractility, intracellular calcium, and repolarization of the heart using well known physiology and electrophysiology techniques. The authors also propose a mechanism for the repolarization abnormalities involving a paraquat mediated effects of outward potassium current kinetics and gating (Ito).

Although this study provides some insight into a mechanism that may mediate the cardiotoxic effects of paraquat, there key concerns that must be addressed before consideration of publication.

1) There are numerous grammatical mistakes within the text. Consider utilizing a proofreading service to identify and correct the grammar before resubmitting.

2) There are multiple studies that have investigated the cellular mechanisms of paraquat poisoning in the heart of rodent models. In fact, several studies have previously reported deficits to cardiomyocyte contractility and intracellular calcium levels (Dong XS et al. 2014; Zhang L et al. 2018; Wang J et al., 2015). Please provide a more comprehensive elaboratation on the previously published findings of paraquat cardiac toxicity and how the findings in your present study are novel, support or differ from what has previously been reported in the introduction and/or discussion sections.

3) The authors used a dosage of 100-180mg/kg of paraquat for the in vivo studies and a concentration of 1-60µM paraquat in vitro. Please provide a clear justification for the use of different concentrations in different studies and how they can be compared. Also, how do the chosen concentrations of paraquat relate to human paraquat poisoning cases.

4) Did any mice did after receiving the paraquat dose in vivo. It would be important to mention the mortality rate in your in vivo injections and if changes to QTc are associated with earlier/later mortality.

5) It should be discussed why it reduced HR. Is it likely it has an effect on the SA node and funny current or depolarization/repolarization time? Did it cause any SA/AV node block?

6) Why was the standard Bazett correction chosen to normalize the QT interval to heart rate in this study? It is known that QTc is over and under estimated at high and low heart rates, respectively. This overestimation is apparent in data presented in table 1 where the average QTc intervals are essentially the same length as the average RR intervals. A normalized Bazett formula, which normalizes QT to average rat RR interval (Kmecova J et al. 2010), has been applied successfully in rodent QT interval correction.

7) Why was QTc the only ECG wave parameter reported in table 1? Based on the raw ECG traces in Figure 1, there seems to be visible changes occurring to more than QT interval such as changes to the P wave and QRS wave. Increased QRS duration also contributes to QT interval so it would be important to know if there is a change in the QRS complex occurring and would add to the novelty of the data.

8) For the studies depicted in figure 3, the same cells were used for control and paraquat administration. Could the parameters have declined naturally over time? Were any controls run for the same amount of time to determine what the change in calcium or shortening occurred with time and to compare to the difference in paraquat. It is also not entirely clear what statistical analysis was performed on the cardiac myocyte shortening and calcium imaging data; if the same cells are used as the control group then the appropriate statistical analysis would be a repeated measures ANOVA.

9) It is not entirely clear how the statistics were analyzed and what P values were obtained for the data points in Figure 4, panels B-D.

Minor comments:

1) line 116,: tyrodes solution component are shown in “µM” however it should be “mM”. Please confirm that all usages of “µM” are correct throughout the text.

2) Please indicate incubation times for paraquat within the methods as it is not entirely clear from the results.

3) Was not able to find Table S1 linked with the manuscript submission

4) Supplemental figure S1 is duplicated as figure 4 in the submission.

5) The discussion is repetitive in describing results in some places. For example the same results are described on lines 329-334 as on lines 342-346.

Reviewer #2: This study from Chih-Chuan Lin and colleagues aimed to investigate the mechanisms by which the herbicide paraquat (PQ) causes cardiotoxicity. Investigations performed on rats included in vivo analysis of cardiac hemodynamics, ex vivo Langendorff heart perfusions, and measurements of intracellular Ca2+ transients and K+ currents in isolated cardiomyocytes. The investigators found that PQ decreased cardiac contractility and increased QTc interval, decreased the amplitude of Ca2+ transients and suppressed K+ currents. The experiments were carefully executed, although the novel mechanistic insights gained from those experiments are limited to the inhibition of K+ currents. The role of this novel finding in the cardiotoxic effects of PQ should be better integrated in the available literature on the topic.

1- In the abstract and the introduction, the authors make the point that the mechanism of acute PQ poisoning induced cardiotoxicity is poorly understood. There are however, several studies which previously reported involvement of inflammation and oxidative stress through pathways involving the proteins TLR4, ETA and FoxO3 (see PMID 23969119, PMID 26089164 and PMID 31264342). Those mechanisms should be mentioned in the text (particularly in the introduction which is no covering the current state of knowledge the way it is presently written). In addition, the discussion should elaborate on how the new mechanistic findings (inhibition of K+ channels by PQ) relate to those previously reported cardiotoxic mechanisms.

2- Similarly, the authors conclude in their discussion that there is currently no established treatment for PQ poisoning, besides some claims about the potential usefulness of immunosuppressive therapies. There are, however, a few other therapies which have been explored for PQ-induced cardiotoxicity (see for example PMID 31425380 regarding the use of atorvastatin, PMID 31083174 with edaravone, and PMID 19026709 with lysine acetylsalicylate) and these should be cited as well.

3- Although it is somewhat discussed on page 19, could the authors be more specific about the relationship between the PQ doses used for in vivo and ex vivo experiments and the levels reported in cases of human poisoning? In other words, have the doses used for the present experiments been reported in poisoned individuals, thereby reinforcing the clinical relevance of the study?

4- Figure 1: A “Control 3h” should also be presented on that figure.

5- Figure 4 (labeled by mistake as Figure 5 in manuscript): The y axes from the graphs presenting the current traces in panel A should include more values to have a better idea of the amplitudes in each condition depicted. Panels B, C, D, E: Any significant differences between curves and points should be indicated on the graphs.

Minor comments:

1- Line 74: Per journal requirements, the ethics statement on animal research should also indicate that “all efforts were made to minimize animal suffering”.

2- Line 236: Table 1 is referred to as TableS1

3- Line 268: “PQ” is missing from that sentence.

4- Figure S1 also appears in the manuscript as Figure 4, therefore the labeling of figures starting with figure 4 is not accurate in the text.

5- Lines 303-307: This information should be reported in the Methods section only.

6. PLOS authors have the option to publish the peer review history of their article (what does this mean?). If published, this will include your full peer review and any attached files.

Reviewer #1: No

Reviewer #2: No

---

## [Author Response · Author response to Decision Letter 0]

17 Feb 2021

Response to Editor’s comment:

Thank you for submitting "Hemodynamic and electromechanical effects of paraquat in rat heart". Two reviewers have completed review and have raised some valid critiques which should be addressed upon revison. In addition, I would like you to address reasons for the relatively low hemodynamic values observed in the langendorff perfused hearts (perfusion pressure etc...). Also relating to the langendorff experiments, it is not clear if separate untreated control hearts were run for the duration (60 min) or if the shown control is just pre-values, an untreated 60 min control would be ideal. Please clarify this in addition to the reviewers comments in your response. There were some noted strengths of the work identified including the patch clamp analysis and other figures, and we would like to reconsider a revised version of the manuscript.

Authors: First of all, we thank the Editor’s comments, giving us the opportunity to revise the manuscript and implement the study with new experiments. The observed amplitude of cardiac performance parameters obtained from the Langendorff-perfused heart study may depend on various factors such as the conditions of perfusion pressure setting (eg, constant perfusion pressure of 55 mmHg in our study) and predrug equilibration time. To acquire reliable parameters after vehicle or drug treatment for at least 1 hour period, we allowed the preparations to equilibrate for 2-2.5 hours to reach the steady-state LV pressure (LVP) value before the commencement of experiments. This time setting was determined by our predrug equilibration time control experiment. We observed that the initial LVP and its derivative were high at the time following the insertion of balloon cathether and then gradually declined within around 2 hours. The averaged initial value of LVP obtained from 8 hearts was 59.9 ± 11.3 mmHg and the value declined to 55.4 ± 7.4, 45.0 ± 6.3, 37.7 ± 6.8 and 35.6 ± 7.2 mmHg following 1, 1.5, 2 and 2.5 hours observation periods, respectively. The initial value of the maximal rise velocity of LV pressure (LV +dP/dt)max was 2192 ± 481 mmHg/s and declined to 2011 ± 360, 1647 ± 379, 1442 ± 362 and 1407 ± 303 mmHg/s following 1, 1.5, 2 and 2.5 hours, respectively.

Exactly, the control traces in Figure 2A and B and the baseline values presented in the text denote pre-drug waveforms and pre-values, respectively. As your suggestion, a corresponding vehicle and time control study run for 60 min had been performed and the data have been included in the revised manuscript as Supplemental Figure S2. 

Response to Editor’s requirement:

Journal Requirements:

Authors: We are sure that our manuscript meets PLOS ONE's style requirements.

2. Please clarify how many rats were used in the in vivo experiment, and how many were in the control group versus the paraquat group.

Authors: A total of 30 animals were used in the in vivo experiment; 10 animals each in vehicle group, paraquat 100 mg/kg group, and 180 mg/kg group.

3. Please clarify how many rats were used in the in vitro experiment.

Authors: A total of 71 rats were used in the in vitro experiment; 28 rats were used for isolated heart study, 20 rats were used for single myocyte study to measure the intracellular calcium fluorescence and cell shortening, and the remaining (23) were used for single myocyte ionic channel current study.

4. Please carefully proofread your manuscript for typographical errors. For example, “The rats was anesthetized with pentobarbital sodium …” should be the “The rats were anesthetized with pentobarbital sodium …”

Authors: We thank for the editor’s comment and we have carefully proofread the manuscript for typographical and grammatical errors.

5. Please specify in your methods section the method of sacrifice, particularly for the control rats in the in vivo experiment.

Authors: The method for sacrifice of the animals (euthanasia) has been specified in the Method section (Lines 144-146) in this revised version. Animals were sacrificed by cervical dislocation following deep anesthesia with overdose anesthetic (1.25 g/kg urethane by intravenous route).

6. Please ensure that you refer to Figure 6 in your text as, if accepted, production will need this reference to link the reader to the figure.

 Authors: We apologize that in fact there was no Figure 6 in the manuscript because of the repetitive positing of Figure S1 as Figure 4.

7. Please include a caption for figure 6.

Authors: We apologize that in fact there was no Figure 6 in the manuscript because of the repetitive positing of Figure S1 as Figure 4.

Response to reviewers’ comment

Reviewer #1: 

1) There are numerous grammatical mistakes within the text. Consider utilizing a proofreading service to identify and correct the grammar before resubmitting.

Authors: We thank for the editor’s comment and we have carefully proofread the manuscript for typographical and grammatical errors.

2) There are multiple studies that have investigated the cellular mechanisms of paraquat poisoning in the heart of rodent models. In fact, several studies have previously reported deficits to cardiomyocyte contractility and intracellular calcium levels (Dong XS et al. 2014; Zhang L et al. 2018; Wang J et al., 2015). Please provide a more comprehensive elaboration on the previously published findings of paraquat cardiac toxicity and how the findings in your present study are novel, support or differ from what has previously been reported in the introduction and/or discussion sections.

Authors: We agree to the reviewer’s excellent points that we should comprehensively compare the results of the PQ exposure on cardiomyocyte contractility and intracellular calcium handling in our study with those previously reported findings as mentioned by the reviewer. We have improved the discussion and the description has been added to the Discussion section in Lines 435-458 in this revised manuscript.

3) The authors used a dosage of 100-180mg/kg of paraquat for the in vivo studies and a concentration of 1-60 µM paraquat in vitro. Please provide a clear justification for the use of different concentrations in different studies and how they can be compared. Also, how do the chosen concentrations of paraquat relate to human paraquat poisoning cases.

Authors: According to the ingested PQ dose, the severity of PQ poisoning in human are classified as mild, moderate to severe, and massive poisoning. Therefore, we chose different PQ doses to simulate the clinical situations as human poisoning. The explanation for the reason why we used different concentrations in different experiments is depicted in the Method section (Lines 103-116) and Discussion section (Lines 392-404).

In humans, ingestion of dosages larger than 40 mg/kg BW may cause acute fulminant poisoning with extremely high death rate (Vale et al., 1987). In general, the lethal dose in rodents is much higher than that in humans. Therefore, in the present animal study, we chose the dosages several fold to that in humans which may cause massive poisioning to simulate the manifestations in clinical PQ poisoning.

The dosages of 100 and 180 mg/kg PQ in vivo should produce approximate blood concentrations of 5.56 mM and 10 mM, respectively, assume the blood volume is approximate to 7% of the body weight. However, the estimated concentrations cannot directly be applied to the in vitro study unless with proper adjustment. This is because the action of drug is confined in the heart and also the drug can easily access to it target site on the cell in the in vitro conditions. While in the in vivo conditions, the action of drug may be affected by multiple factors such as the pharmacokinetic property, availability of the drug, diffusion barrier, and many other factors. Thus, we used much lower concentrations (eg, 1-60 µM) than the estimated ones for our in vitro study.

4) Did any mice die after receiving the paraquat dose in vivo. It would be important to mention the mortality rate in your in vivo injections and if changes to QTc are associated with earlier/later mortality.

Authors: Many animals in PQ receiving group died at the later stage of the observation period while all animals in vehicle-treated group survived till the end of the experiment. We calculated the mortality rate and found that the mortality rate of 100 mg/kg and 180 mg/kg PQ-treated rats was 60% (6/10) and 90% (9/10), respectively. We also found that the longer QTc interval is correlated with the higher and early mortality rate following PQ treatment. These data were added to the Result section and described in Lines 285-289.

5) It should be discussed why it reduced HR. Is it likely it has an effect on the SA node and funny current or depolarization/repolarization time? Did it cause any SA/AV node block?

Authors: Thanks to the Reviewer’s excellent observation. We noted that paraquat administration decreased heart rate in anesthetized rats. It is known that any decreases in the depolarizing currents such as funny channel current (If), T-type or L-type calcium current, or the repolarizing current such as delayed outward potassium current of the SA node may delay the depolarization or repolarization time, respectively, and contribute to slowing of heart rate. We did not examine these ionic channels in the present study. Further studies are needed to clarify whether PQ could suppress the sinus nodal depolarization or repolarization currents and thereby decreasing the heart rate. This discussion and limitation has been described in the Discussion section (Lines 410-419). As paraquat administration also prolonged the P wave duration and P-R interval, which denote SA nodal-atrial depolarization time and atrial-ventricular (AV nodal) conduction time, respectively, at the later stage of the observation period, it is possible that some mortality of the animals could be due to the severe SA or AV nodal blockade.

6) Why was the standard Bazett correction chosen to normalize the QT interval to heart rate in this study? It is known that QTc is over and underestimated at high and low heart rates, respectively. This overestimation is apparent in data presented in table 1 where the average QTc intervals are essentially the same length as the average RR intervals. A normalized Bazett formula, which normalizes QT to average rat RR interval (Kmecova J et al. 2010, QTcn-B = QT / (RR / f)1/2, f = 150 ms ), has been applied successfully in rodent QT interval correction.

Authors: Thanks for the reviewer’s comment. We referred to the reference mentioned by the reviewer and we agree to the reviewer’s comment that the normalized Bazett’s formula should be more suitable than the conventional one for the correction of QT interval in rodent model. We have recalculated all the QTc values in each group by using the normalized Bazett’s formula and therefore the QTc data in Table 1 were renewed in this revised version.

7) Why was QTc the only ECG wave parameter reported in table 1? Based on the raw ECG traces in Figure 1, there seems to be visible changes occurring to more than QT interval such as changes to the P wave and QRS wave. Increased QRS duration also contributes to QT interval so it would be important to know if there is a change in the QRS complex occurring and would add to the novelty of the data.

Authors: Thanks for the reviewer’s comment. We agree to the reviewer’s comment that the parameters other than QTc interval may also be changed following paraquat administration in vivo. Thus we reanalyzed the ECG data and the parameters of P wave duration and PR and QRS intervals were added to Table 1 in this revised manuscript. Besides, we apologize that we omitted the QT interval data in Table 1 in our previous version. These data were added to Table 1 in this revised manuscript.

8) For the studies depicted in figure 3, the same cells were used for control and paraquat administration. Could the parameters have declined naturally over time? Were any controls run for the same amount of time to determine what the change in calcium or shortening occurred with time and to compare to the difference in paraquat. It is also not entirely clear what statistical analysis was performed on the cardiac myocyte shortening and calcium imaging data; if the same cells are used as the control group then the appropriate statistical analysis would be a repeated measures ANOVA.

Authors: We used the same cells both in the predrug control and drug treatment conditions, therefore, p values in Figure 3C and 3D were conducted by GEE analysis to evaluate the changes of variables over time. This description was added to the Statistic subsection under the Materials and Methods section in Lines 256-258).

9) It is not entirely clear how the statistics were analyzed and what P values were obtained for the data points in Figure 4, panels B-D.

Authors: We are sorry that we misplace the supplementary Fig. S1 as Fig 4 in the previous submission. Therefore, the statistics used in the previous submission was GEE as that for fig. 3C/3D.

Minor comments:

1) line 116,: tyrodes solution component are shown in “µM” however it should be “mM”. Please confirm that all usages of “µM” are correct throughout the text.

Authors: We apologize for the typographical error. All the units including the composition of the solution and calcium and cobalt ions previously labeled with “µM” (Line 116,137, 142, 144, 149, 151, 152, 184, 199, 273 in previous version) were corrected as “mM” throughout the Method and Results sections in the present version.

2) Please indicate incubation times for paraquat within the methods as it is not entirely clear from the results.

Authors: The incubation time of paraquat for all the single myocyte studies was around 4.5 minutes. This description has been added to the Methods section (Lines 210, 248-249).

3) Was not able to find Table S1 linked with the manuscript submission.

Authors: We apologize that Table S1 was not uploaded successfully in our previous submission. We will confirm the uploading in this revised version.

4) Supplemental figure S1 is duplicated as figure 4 in the submission.

Authors: Thank you for reminding us this error. We have deleted the duplicated Figure 4 and made sure all the Figures were accurate.

5) The discussion is repetitive in describing results in some places. For example the same results are described on lines 329-334 as on lines 342-346.

Authors: According the reviewer’s comment, the redundant descriptions (Lines 342-347 in previous version) in Discussion section were deleted.

Reviewer #2: 

This study from Chih-Chuan Lin and colleagues aimed to investigate the mechanisms by which the herbicide paraquat (PQ) causes cardiotoxicity. Investigations performed on rats included in vivo analysis of cardiac hemodynamics, ex vivo Langendorff heart perfusions, and measurements of intracellular Ca2+ transients and K+ currents in isolated cardiomyocytes. The investigators found that PQ decreased cardiac contractility and increased QTc interval, decreased the amplitude of Ca2+ transients and suppressed K+ currents. The experiments were carefully executed, although the novel mechanistic insights gained from those experiments are limited to the inhibition of K+ currents. The role of this novel finding in the cardiotoxic effects of PQ should be better integrated in the available literature on the topic.

Authors: First of all, we thank the Reviewer’s comments, giving us the opportunity to revise the manuscript and implement the study with new experiments.

1- In the abstract and the introduction, the authors make the point that the mechanism of acute PQ poisoning induced cardiotoxicity is poorly understood. There are however, several studies which previously reported involvement of inflammation and oxidative stress through pathways involving the proteins TLR4, ETA and FoxO3 (see PMID 23969119, PMID 26089164 and PMID 31264342). Those mechanisms should be mentioned in the text (particularly in the introduction which is no covering the current state of knowledge the way it is presently written). In addition, the discussion should elaborate on how the new mechanistic findings (inhibition of K+ channels by PQ) relate to those previously reported cardiotoxic mechanisms.

Authors: We agree with the Reviewer’s comments that we should cover the known updated studies on the possible underlying mechanism of PQ-induced cardiotoxicity in our manuscript. According the reviewer’s comments, the suggested 3 references were cited and their main findings were described in the Abstract and Introduction sections (Lines 22-24, 66-72). As these 3 references did not provide information about the electrical abnormalities or related mechanisms following PQ administration, while our present study defined the electrical basis for the PQ-induced QTc prolongation. The descriptions were added in the Discussion section to emphasize our novel findings (Lines 437-443, 456-458).

2- Similarly, the authors conclude in their discussion that there is currently no established treatment for PQ poisoning, besides some claims about the potential usefulness of immunosuppressive therapies. There are, however, a few other therapies which have been explored for PQ-induced cardiotoxicity (see for example PMID 31425380 regarding the use of atorvastatin, PMID 31083174 with edaravone, and PMID 19026709 with lysine acetylsalicylate) and these should be cited as well.

Authors: We agree with the Reviewer’s comments that we should cover the other updated therapeutic strategies which have been explored for PQ-induced cardiotoxicity in our manuscript. According the reviewer’s comments, the suggested references were cited and their main findings were described in the Discussion section (Lines 494-504).

3- Although it is somewhat discussed on page 19, could the authors be more specific about the relationship between the PQ doses used for in vivo and ex vivo experiments and the levels reported in cases of human poisoning? In other words, have the doses used for the present experiments been reported in poisoned individuals, thereby reinforcing the clinical relevance of the study?

Authors: The reported LD50 value in rats was around 57 mg/kg. In this study, we used 100 and 180 mg/kg (approximately twice and triple LD50 value, respectively) for the in vivo study. In humans, ingestion of dosages larger than 40 mg/kg BW may cause acute fulminant poisoning with extremely high death rate (Vale et al., 1987). In general, the lethal dose in rodents is much higher than that in humans. Therefore, in the present animal study, we chose the dosages several fold to that in humans which may cause massive poisioning to simulate the manifestations in clinical PQ poisoning.

For the in vitro study, the concentrations used were estimated according to the calculated blood concentrations of PQ (Mw 257.16), assume that the blood volume (mL) is estimated to be 7% of the body weight (g). Consequently, dosages of 100 and 180 mg/kg PQ may produce approximate blood concentrations of 5.56 mM and 10 mM, respectively. However, the estimated concentrations cannot directly be applied to the in vitro study unless with proper adjustment. This is because the action of drug is confined in the heart and also the drug can easily access to it target site on the cell in the in vitro conditions. In contrast, in the in vivo conditions, the action of drug may be affected by multiple factors such as the pharmacokinetic property, availability of the drug, diffusion barrier, and many other factors. Thus, we used much lower concentrations (eg, 1-60 µM) than the estimated ones for our in vitro study. Although the intoxication dosages in rats are much higher than those in humans, our study showed that the dose-dependent cardiac suppressive effects of PQ are comparable to the clinical conditions in PQ poisoning cases. The descriptions were added to the Method section (Lines 103-116) and Discussion section (Lines 392-404).

4- Figure 1: A “Control 3h” should also be presented on that figure.

Authors: According to the reviewer’s comments, the representative tracings of the saline vehicle and time control group of in vivo study were prepared and provided as Supplemental Figure S1.

5- Figure 4 (labeled by mistake as Figure 5 in manuscript): The y axes from the graphs presenting the current traces in panel A should include more values to have a better idea of the amplitudes in each condition depicted. Panels B, C, D, E: Any significant differences between curves and points should be indicated on the graphs.

Authors: According to the reviewer’s comments, more values for labeling of the y axis in panel A of Figure 4 were added to improve the presentation quality. In panel B-D, no labeling to denote the significant differences are shown because there were trend of changes between curves and points following paraquat treatment but the differences did not reach statistical significance.

Minor comments:

1- Line 74: Per journal requirements, the ethics statement on animal research should also indicate that “all efforts were made to minimize animal suffering”.

Authors: The ethics statement “all efforts were made to minimize animal suffering” was added in Line 95 in the Material and Methods section.

2- Line 236: Table 1 is referred to as Table S1

Authors: We are sure that Table S1 here is the correct one as this Table denotes the predrug basline values of the hemodynamic and ECG data.

3- Line 268: “PQ” is missing from that sentence.

Authors: We thank for the reviewer’s observation. In deed, “PQ” is missing from the sentence. As the comment of the results was also depicted in the Discussion section, this sentence was therefore deleted in this revised manuscript.

4- Figure S1 also appears in the manuscript as Figure 4, therefore the labeling of figures starting with figure 4 is not accurate in the text.

Authors: We apologize for the error of positing Figure S1 as Figure 4. We rechecked all the figures and made sure the labeling were correct throughout the text in the revised version.

5- Lines 303-307: This information should be reported in the Methods section only.

Authors: Thank for the reviewer’s suggestion and this description was moved to the Material and Methods section (Lines 240-243) in this revised version.

---

## [Decision Letter · Decision Letter 1]

19 Mar 2021

Hemodynamic and electromechanical effects of paraquat in rat heart

PONE-D-20-15928R1

Dear Dr. Chang,

We’re pleased to inform you that your manuscript has been judged scientifically suitable for publication and will be formally accepted for publication once it meets all outstanding technical requirements.

Kind regards,

Richard T. Clements, PhD

Academic Editor

PLOS ONE

Additional Editor Comments (optional):

Please also make the grammatical revisions requested by reviewer 2. 

Reviewers' comments:

Reviewer's Responses to Questions

**Comments to the Author**

1. If the authors have adequately addressed your comments raised in a previous round of review and you feel that this manuscript is now acceptable for publication, you may indicate that here to bypass the “Comments to the Author” section, enter your conflict of interest statement in the “Confidential to Editor” section, and submit your "Accept" recommendation.

Reviewer #1: All comments have been addressed

Reviewer #2: All comments have been addressed

2. Is the manuscript technically sound, and do the data support the conclusions?

Reviewer #1: (No Response)

Reviewer #2: Yes

3. Has the statistical analysis been performed appropriately and rigorously? 

Reviewer #1: (No Response)

Reviewer #2: Yes

4. Have the authors made all data underlying the findings in their manuscript fully available?

Reviewer #1: (No Response)

Reviewer #2: Yes

5. Is the manuscript presented in an intelligible fashion and written in standard English?

Reviewer #1: (No Response)

Reviewer #2: No

6. Review Comments to the Author

Reviewer #1: (No Response)

Reviewer #2: All my concerns have been addressed adequately. Please proofread the manuscript carefully one more time as typographical errors are still found throughout the text, and especially in the introduction. For example:

- Sentence at lines 54-55 is not clear

- Line 56: "mostly or one week"? This part of the sentence is not clear

- Line 56: "injured"

- Line 61: "may lead to triggered activity" . This part of sentence is not clear

- Line 111: "assuming that blood volume (in mL) represents 7% of the body weight (in g)."

- Line 171: "at a rate of 6 mL/min/g cardiac tissue"

- Line 215: "A small aliquot of dissociated cells was placed..."

- Line 440: Review this sentence which is no clear and provide meaning of ETA abbreviation (endothelin receptor type A) in sentence.

7. PLOS authors have the option to publish the peer review history of their article (what does this mean?). If published, this will include your full peer review and any attached files.

Reviewer #1: No

Reviewer #2: No

---

## [Editor Report · Acceptance letter]

25 Mar 2021

PONE-D-20-15928R1 

Hemodynamic and electromechanical effects of paraquat in rat heart 

Dear Dr. Chang:

I'm pleased to inform you that your manuscript has been deemed suitable for publication in PLOS ONE. Congratulations! Your manuscript is now with our production department. 

Kind regards, 

on behalf of

Dr. Richard T. Clements 

Academic Editor

PLOS ONE